# CDYL suppresses epileptogenesis in mice through repression of axonal Nav1.6 sodium channel expression

Yongqing Liu[1,2], Shirong Lai[1,2], Weining Ma[3], Wei Ke[4], Chan Zhang[5], Shumeng Liu[1], Yu Zhang[1], Fei Pei[6], Shaoyi Li[3], Ming Yi[5], Yousheng Shu[4], Yongfeng Shang[1,7,8], Jing Liang[1] & Zhuo Huang[1,2,9]

Impairment of intrinsic plasticity is involved in a range of neurological disorders such as epilepsy. However, how intrinsic excitability is regulated is still not fully understood. Here we report that the epigenetic factor Chromodomain Y-like (CDYL) protein is a critical regulator of the initiation and maintenance of intrinsic neuroplasticity by regulating voltage-gated ion channels in mouse brains. CDYL binds to a regulatory element in the intron region of SCN8A and mainly recruits H3K27me3 activity for transcriptional repression of the gene. Knockdown of CDYL in hippocampal neurons results in augmented Nav1.6 currents, lower neuronal threshold, and increased seizure susceptibility, whereas transgenic mice over-expressing CDYL exhibit higher neuronal threshold and are less prone to epileptogenesis. Finally, examination of human brain tissues reveals decreased CDYL and increased SCN8A in the temporal lobe epilepsy group. Together, our findings indicate CDYL is a critical player for experience-dependent gene regulation in controlling intrinsic excitability.

[1] Key Laboratory of Carcinogenesis and Translational Research (Ministry of Education), State Key Laboratory of Natural and Biomimetic Drugs, Beijing Key Laboratory of Protein Posttranslational Modifications and Cell Function, Department of Biochemistry and Molecular Biology, School of Basic Medical Sciences, Peking University Health Science Center, Beijing 100191, China. [2] Department of Molecular and Cellular Pharmacology, School of Pharmaceutical Sciences, Peking University Health Science Center, Beijing 100191, China. [3] Department of Neurology, Shengjing Hospital Affiliated to China Medical University, Shenyang 110000, China. [4] State Key Laboratory of Cognitive Neuroscience and Learning and IDG/McGovern Institute for Brain Research, School of Brain and Cognitive Sciences, the Collaborative Innovation Center for Brain Science, Beijing Normal University, Beijing 100875, China. [5] Neuroscience Research Institute & Department of Neurobiology, Peking University Health Science Center, Beijing 100191, China. [6] Department of Pathology, Peking University Health Science Center, Beijing 100191, China. [7] Department of Biochemistry and Molecular Biology, School of Basic Medical Sciences, Capital Medical University, Beijing 100069, China. [8] Department of Biochemistry and Molecular Biology, Tianjin Medical University, Tianjin 300070, China. [9] State Key Laboratory of Organometallic Chemistry, Chinese Academy of Sciences, Shanghai 200032, China. Yongqing Liu and Shirong Lai contributed equally to this work. Correspondence and requests for materials should be addressed to J.L. (email: liang_jing@bjmu.edu.cn) or to Z.H. (email: huangz@hsc.pku.edu.cn)

Sensory experience or spontaneous internal stimuli continuously refine neuronal network activity in mammalian brain to promote learning and memory[1, 2]. The ability of a neuron to change in excitability over time, namely neuroplasticity, is the fundamental neural basis of behavioral change. There are two broadly recognized categories of activity-dependent plasticity: synaptic and nonsynaptic. Synaptic plasticity, also referred to as Hebbian plasticity, deals directly with the strength of synapses between neurons. By contrast, nonsynaptic plasticity involves modification of neuronal excitability in the axon, dendrites, and soma of a single neuron[3]. Although it is generally

recognized that Hebbian plasticity is critical in behavior-modifying changes in neuronal connectivity, emerging evidence suggests intrinsic plasticity also strongly influences neuronal network activity[3, 4].

One important function of intrinsic plasticity is to shape the input-output information flow from dendrites to axon terminals, through modulating expression levels or biophysical properties of various ion channels localized to different neuronal compartments[4–6]. Earlier studies demonstrated that in response to neuronal activity, changes in dendritic HCN1 channels or Kv4 potassium channels modify integration and propagation of

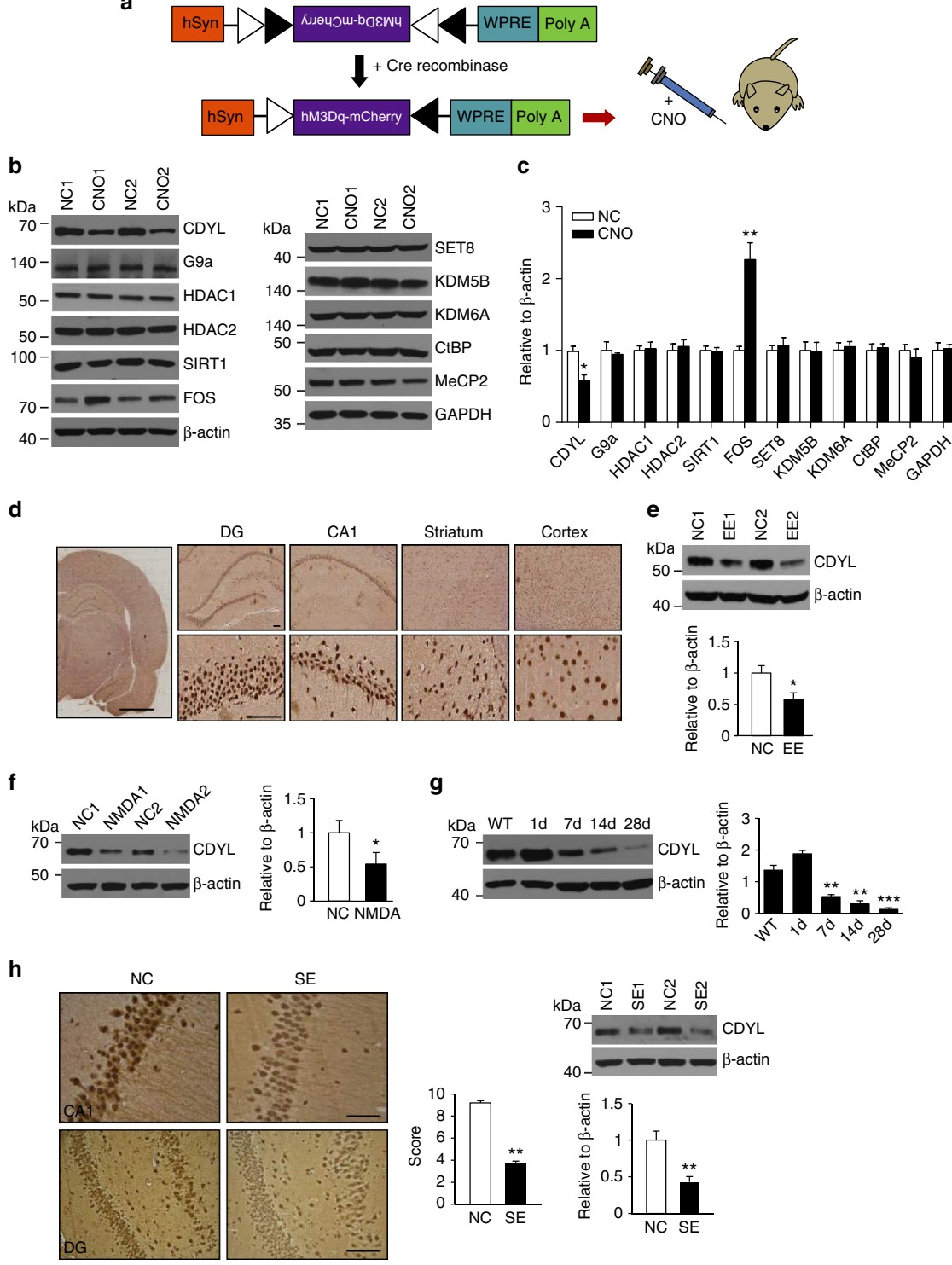

synaptic inputs to the soma[7, 8], whereas sodium channels localized to axonal initial segment (AIS) regulate action potential (AP) initiation and backpropagation[6, 9, 10]. Once induced, experience-related changes in intrinsic plasticity can last days or months[11, 12]. Therefore, intrinsic plasticity is considered to be an important component and modulator of learning and memory[13]. Furthermore, impairment of intrinsic plasticity is involved in a range of neurological and psychiatric disorders such as addiction, pain, and epilepsy[14, 15]. Despite the importance of intrinsic neuroplasticity in physiological and pathological processes in the brain, the underlying molecular mechanisms are still poorly understood.

The induction and maintenance of long-term plasticity require alteration of gene transcription in neuronal cells[16]. Epigenetic regulation is an important means to alter cell-wide gene expression in response to various environmental stimuli, leaving epigenetic factors to be ideal candidates for modulating intrinsic plasticity[17, 18]. Post-translational modification of histones, such as acetylation, methylation, phosphorylation, and ubiquitination, is an important epigenetic mechanism for gene transcriptional regulation[19, 20]. Various enzymes add or remove specific modifications at specific histone residues, and the embedded information is further transmitted by "reader" proteins, which recognize the signal and subsequently recruit downstream transcriptional regulators to tune up or tune down the gene expression[21–23].

Here, we report chromodomain Y-like (CDYL) protein, a histone methyllysine reader and transcriptional corepressor, is critically involved in the induction and maintenance of activity-dependent intrinsic plasticity. CDYL expression is decreased upon enhanced network activity. Genome-wide chromatin immunoprecipitation (ChIP)-sequencing analysis reveals that CDYL regulates multiple neuronal functional pathways, including voltage-gated ion channels, of which SCN8A is the major target gene responsible for the function of CDYL in regulating AP threshold. We show CDYL binds to a regulatory element in the intron region of SCN8A gene, and further recruits mainly the H3K27me3 activity to repress SCN8A transcription in both mouse brains and human SY5Y cells. Knockdown of CDYL in mouse hippocampal neurons results in augmented Nav1.6-mediated sodium currents, increased seizure susceptibility and decreased latent period duration. By contrast, compared to wild-type littermates, transgenic mice over-expressing CDYL exhibit reduced Nav1.6 function and are less prone to epileptogenesis. Finally, examination of human temporal lobe epilepsy (TLE) brain tissues reveals decreased expression of CDYL and increased expression of SCN8A, supporting the important role of CDYL in suppressing epileptogenesis. Together, our findings provide a molecular link between network excitation and neuronal intrinsic excitability, and suggest activity-dependent epigenetic regulation is an important mechanism in controlling behavior changes during physiological and pathological processes of the brain.

## Results

**CDYL is downregulated by neuronal activity in vivo**. Regulated by neuronal activity, intrinsic plasticity takes seconds to occur and can last days or more, during which process gene expression profiles change and epigenetic mechanisms could play a key role[11, 12]. To test the hypothesis, we sought to identify differentially expressed epigenetic factors following enhanced network activities in a DREADD (Designer Receptors Exclusively Activated by Designer Drugs)-based mouse model. To selectively activate hippocampal network, AAV-hM3Dq-mCherry (adeno-associated virus capable of expressing hM3Dq-mCherry in a Cre-dependent manner) was administered into the hippocampus of 10-week-old Camk2a-cre mice that restrictedly expressed Cre recombinase in hippocampal excitatory neurons. The mice were intraperitoneally administered vehicle or clozapine-N-oxide (CNO), which stimulates hM3Dq, to depolarize hippocampal neurons (Fig. 1a)[24, 25]. To confirm that this system enhanced neuronal excitability of hippocampal cells, whole-cell current-clamp recordings were obtained from hippocampal dentate gyrus (DG) granule cells. Bath application of 500 nM CNO to hippocampal slices depolarized DG granule cells by $5.869 \pm 0.797$ mV and increased firing rate of DG granule cells (Supplementary Fig. 1a, b). After 5 days of CNO treatment, mouse hippocampal tissues were collected, lysed, and western blot analysis was performed with antibodies against a panel of epigenetic factors (Fig. 1b, c). The induction of c-fos was included as a marker for neuronal activation[25]. Upon CNO administration, levels of most of the epigenetic factors we examined were not changed. Interestingly, however, we found the expression of CDYL, which we and others have previously shown is a histone methyllysine reader protein and a transcriptional corepressor[26–29], was significantly decreased when mice exhibited enhanced network activities (Fig. 1b, c).

We subsequently assessed CDYL expression in adult rat brains in order to further investigate the role of CDYL in regulating neuronal excitability in different regions. Immunohistochemical examination showed clear nuclear staining of CDYL in hippocampal DG, CA1, striatum, and cortex, with strong expression in the hippocampal DG and CA1 (Fig. 1d), supporting that CDYL could mainly function in these cells. We next challenged rats with various stimuli to increase network excitability, and examined whether the expression of CDYL would change. Western blotting showed that compared to the rats in standard housing, long time exposure of rats to an enriched environment resulted in decreased CDYL expression in the hippocampus (Fig. 1e). The level of CDYL was also decreased 3 days after NMDA injection to the rat brain (Fig. 1f). In addition, triggering seizures in rats by intraperitoneal injection of

**Fig. 1** CDYL protein is downregulated by neuronal activity in vivo. **a** Illustration of the stimulatory hM3Dq DREADD system. A schematic diagram showed the stimulatory hM3Dq DREADD that is activated by clozapine-N-oxide (CNO). **b** Western blot analysis of indicated proteins in hippocampus tissues obtained from mice injected vehicle or CNO for 5 days ($n = 4$ mice in each group). **c** Quantification of the western blotting in **b** by normalizing the levels of the indicated proteins to that of β-actin. $n = 4$, *$p < 0.05$, **$p < 0.01$, paired two-tailed Student's $t$-test. Data were represented as mean ± s.e.m. **d** Immunohistochemistry (IHC) images showing the expression of CDYL protein in DG, CA1, striatum and neocortex area of the rat brain. *Scale bar*, 5 mm (*left*), 0.2 mm (*right*). **e** Western blotting and quantification showing CDYL protein was decreased upon exposure of rats to enriched enviroment (EE). $n = 5$, *$p < 0.05$, paired two-tailed Student's $t$-test. **f** Western blotting and quantification showing intracranial injection of NMDA (10 μM) into rat hippocampus DG area significantly decreased CDYL protein level. $n = 4$, *$p < 0.05$, paired two-tailed Student's $t$-test. **g** Western blotting and quantification showing CDYL expression levels were gradually reduced after status epilepticus in rat temporal lobe epilepsy model. $n = 3$, **$p < 0.01$, ***$p < 0.001$, one-way ANOVA with Bonferroni's multiple-comparisons test. Data marked asterisks in the figure are significantly different from control. **h** IHC results showing CDYL protein was decreased in the 28 days after status epilepticus in rat temporal lobe epilepsy model (*left*). *Scale bar*, 0.1 mm (*upper*), 0.2 mm (*bottom*). The reduction of CDYL expression was confirmed by western blotting and quantification (*right*). $n = 3$, **$p < 0.01$, paired two-tailed Student's $t$-test. Data were represented as mean ± s.e.m.

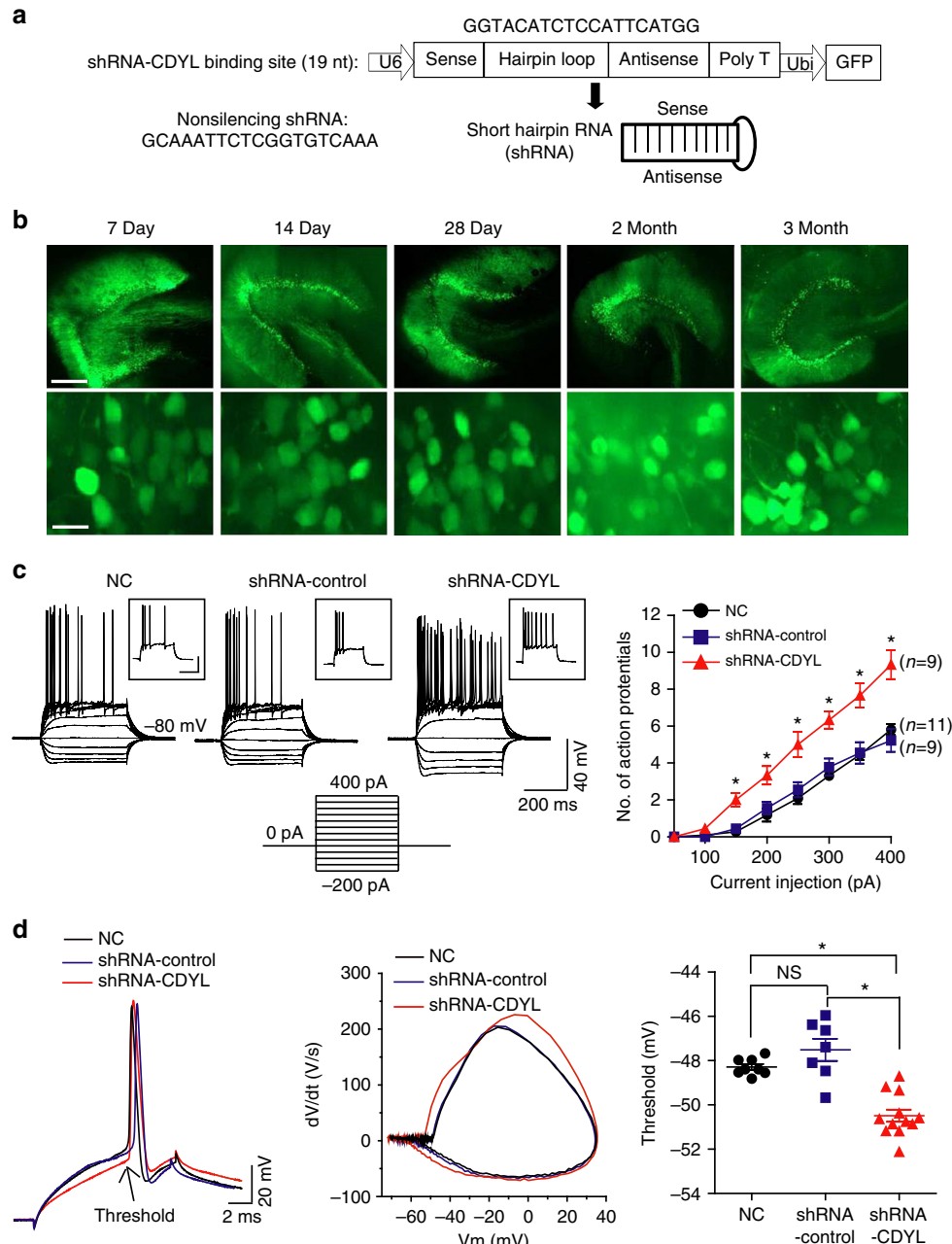

**Fig. 2** Knockdown of CDYL significantly enhanced neuronal intrinsic excitability. **a** A schematic diagram showing the lentiviral shRNA expression vector system. The 19 nuleotide shRNA is specific to the rat/mouse CDYL mRNA. **b** Expression of lentivirus over time in the dorsal hippocampal DG area. Rats infused with lentivirus were transcardially perfused on day 7, 14, 28, 60, 90 after infusion (days post infusion, DPI). Scale bar, 500 μm (upper), 15 μm (bottom). **c** Representative current-clamp recordings obtained from normal control (NC), CDYL-shRNA-infected and nonsilencing-shRNA-infected rat DG neurons (left). A series of 400 ms hyperpolarizing and depolarizing steps in 50 pA increments were applied to produce the traces. For comparison reasons, the cells were held at −80 mV of membrane potential. The mean number of action potentials generated in response of depolarizing current pulses were shown in the right panel. *$p < 0.05$, one-way ANOVA with Bonferroni's multiple-comparison test. **d** Typical spikes (left), associated phase plane plots (middle) and the average spike threshold (right) obtained from control, CDYL-shRNA-infected and nonsilencing-shRNA-infected rat DG neurons. *$p < 0.05$, one-way ANOVA with Bonferroni's multiple-comparison test. Data were represented as mean ± s.e.m.

kainic acid (KA) leads to stepwise decrease of CDYL 7–28 days post-treatment (Fig. 1g). Immunohistochemical staining further confirmed the decreased expression of CDYL in hippocampal DG and CA1 cells 28 days after KA injection (Fig. 1h). Interestingly, unlike CDYL protein, the mRNA level of CDYL did not significantly change upon these challenges (Supplementary Fig. 1c–f), suggesting that protein degradation is likely the underlying mechanism for the diminishment of CDYL in line with our previous study[29]. Together, these results indicate that

CDYL expression is downregulated by neuronal circuit activity, suggesting CDYL could play a role in activity-dependent alteration in neuronal excitability.

**CDYL inhibits intrinsic excitability of hippocampal neurons.** It was previously reported that CDYL knockout (CDYL-KO) mice were embryonic lethal or died shortly after birth[30]. To investigate whether reducing CDYL expression could play a role in eliciting

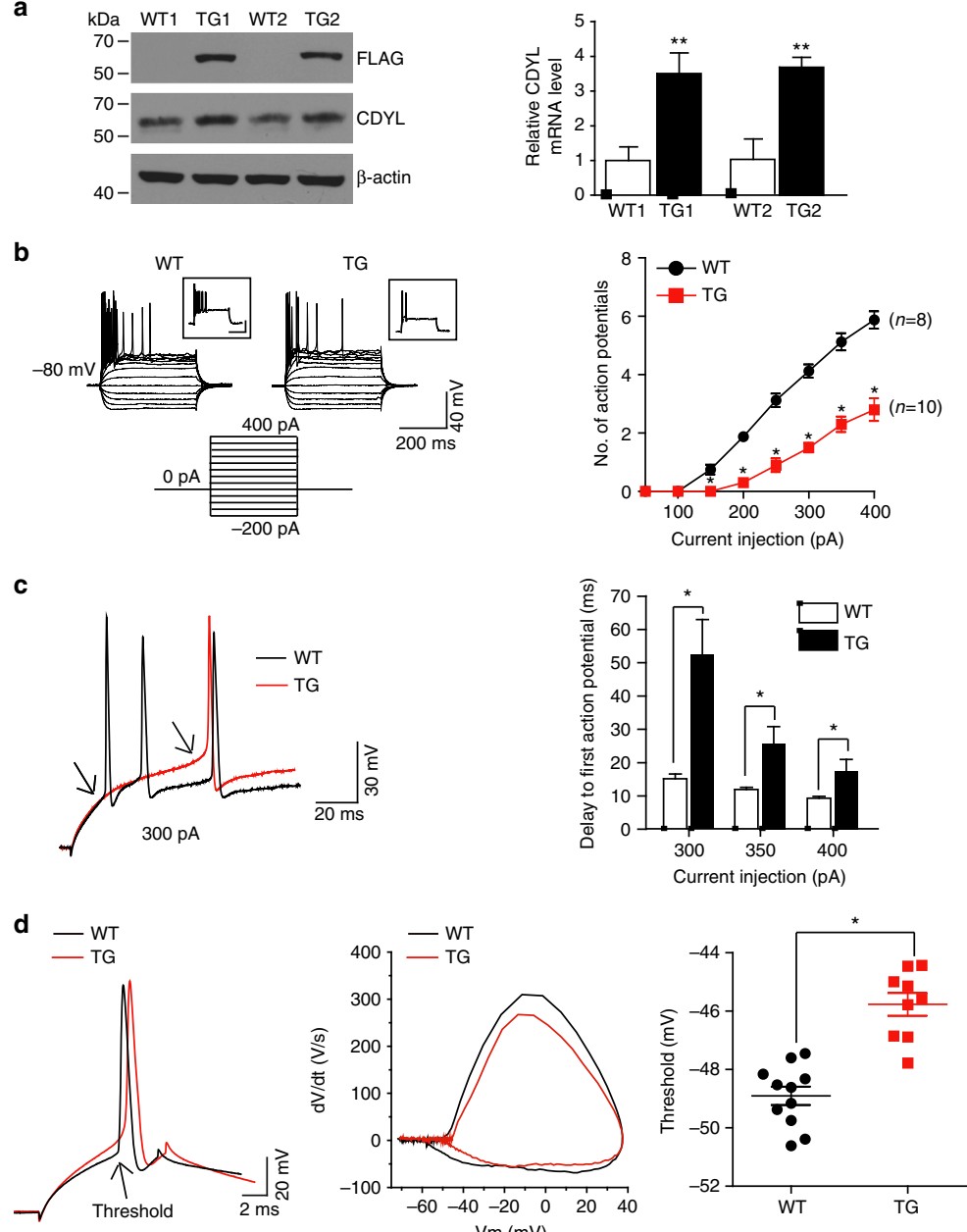

**Fig. 3** Over expression of CDYL in vivo significantly reduced neuronal intrinsic excitability. **a** Western blotting and real-time RT PCR analysis showed CDYL protein and mRNA levels of the progeny from two founders (TG1: #8 and TG2: #24) of transgenic mice over-expressing CDYL. Western blot of FLAG and CDYL in the hippocampus from adult wild-type (WT) and TG littermates (*left*). Quantification of CDYL mRNA levels in hippocampus obtained from WT and TG littermates (*right*). $n = 3$, **$p < 0.01$, paired two-tailed Student's *t*-test. Because of the similar CDYL expression levels in these transgenic mice, which also behaves similarly (Supplemental Fig. 4a–c), we generally used "TG" to represent transgenic mice from both founders in the following functional experiments for simplicity. **b** Representative traces obtained from DG neurons from WT and TG mice in response to a series of 400 ms current stepping from −200 to +400 pA with increments of 50 pA. For comparison reasons, the recordings were obtained at the fixed potential of −80 mV. Graphs demonstrate average numbers of action potentials obtained in response to varying depolarizing current pulses when the soma was held at −80 mV. *$p < 0.05$, one-way ANOVA with Bonferroni's multiple-comparison test. **c** Representative traces of DG neurons from WT and TG mice in response to 300 pA positive current injection (*left*). The avaraged delay of first action potential initiation in WT and TG neurons (*right*). $n = 25$, *$p < 0.05$, unpaired two-tailed Student's *t*-test. **d** Typical spikes (*left*), associated phase plane plots (*middle*), and the average spike threshold (*right*) obtained from DG neurons of WT and TG mice. *$p < 0.05$, unpaired two-tailed Student's *t*-test. Data were represented as mean ± s.e.m.

neuronal excitability in adult mice, we injected lentiviruses carrying control- or CDYL-shRNA-GFP to rat hippocampal DG regions. shRNA-GFP was strongly expressed from 7 days post-infusion (DPI) and lasted at least 3 months (Fig. 2a, b), during which period the viruses spread mediolaterally 1.4–2 mm and anteroposteriorly 1.05–1.75 mm. The efficiency of CDYL silencing was confirmed by western blotting and RT-PCR analysis, which respectively showed 60% reduction of protein and 45% reduction of mRNA of CDYL in virus-infected cells (Supplementary Fig. 2a–f).

We next determined whether knockdown of CDYL had any effect on the intrinsic excitability of hippocampal DG granule

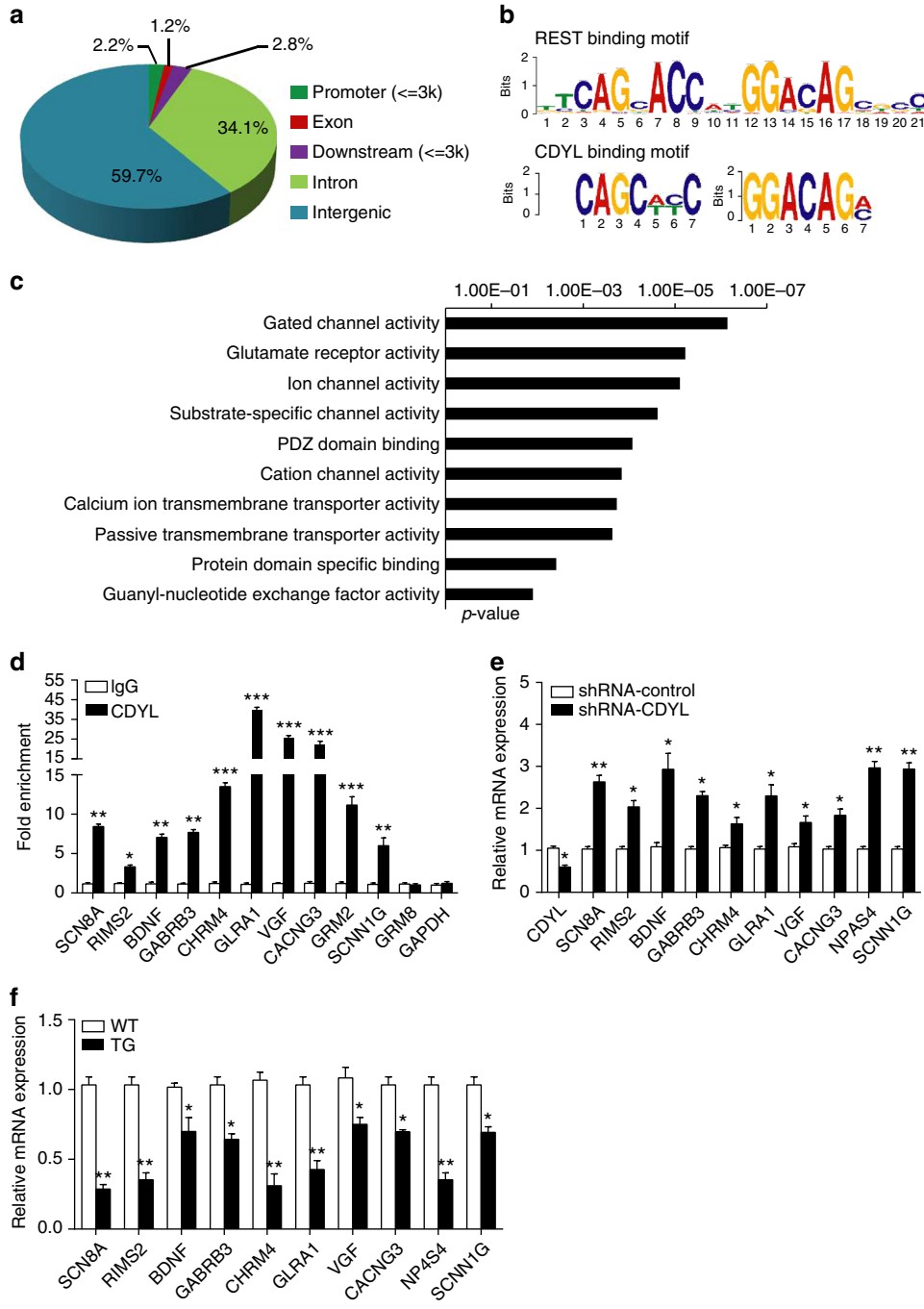

**Fig. 4** Genome-wide identification of CDYL target genes in mouse hippocampus. **a** Genomic distribution of CDYL binding regions determined by ChIP-seq analysis. **b** Consensus CDYL binding motif is nearly identical to the REST binding motif. Motif screening was performed using MEME suite. **c** Classification of the genes identified in ChIP-seq experiments with GO analysis. **d** Verification of the ChIP-seq results in mice hippocampus tissues by ChIP-qPCR. Results are represented as fold change over GAPDH negative control. $n = 3$, *$p < 0.05$, **$p < 0.01$, ***$p < 0.001$, paired two-tailed Student's $t$-test. **e**, **f** Quantitative real-time RT-PCR analysis to measure the mRNA levels of indicated genes. Hippocampus tissues were obtained from mice infected with lentiviruses containing CDYL shRNA or nonsilencing shRNA for 14 days (**e**), or from adult WT or CDYL-TG littermates (**f**). Total RNAs were prepared and mRNA levels of the indicated genes were examined by real-time RT-PCR. The levels of mRNA were normalized against that of GAPDH. *$p < 0.05$, **$p < 0.01$, paired two-tailed Student's $t$-test. Data were represented as mean ± s.e.m.

cells. For this purpose, whole-cell current-clamp recordings were used in rat brain slices infected with control- or CDYL-shRNA on14–21 DPI. The experiments were performed in the presence of inhibitors of glutamate and GABA receptors, unless otherwise stated. Hippocampal DG granule cells were identified by large input resistances (RN) of $166.38 \pm 6.11 (n = 33)$ and complex dendritic trees as revealed by post-hoc morphological analysis.

Neither the morphology (Supplementary Fig. 2g, h) nor resting membrane potentials (RMPs) of neurons were changed with CDYL-shRNA infection (CDYL-shRNA infected neurons RMP, $-77.91 \pm 0.58$, $n = 34$; nonsilencing-shRNA infected neurons RMP, $-77.23 \pm 0.89$, $n = 20$; non-infected neurons RMP, $-78.08 \pm 0.48$, $n = 33$) (Supplementary Table 1). Interestingly, when membrane potentials were held at $-80$ mV with current

injection, significantly increased number of APs was recorded in CDYL-shRNA infected granule cells upon application of depolarizing current pulses (Fig. 2c). Comparable results could also be produced by holding neurons at their normal RMP (Supplementary Fig. 3a, b). In addition, with equal amount of

positive current injection (250–400 pA), CDYL-shRNA infected neurons generated the first AP significantly faster than control cells (Supplementary Fig. 3c). We further performed current-clamp recordings with minimum positive current injection to induce single AP in hippocampal DG neurons. This method

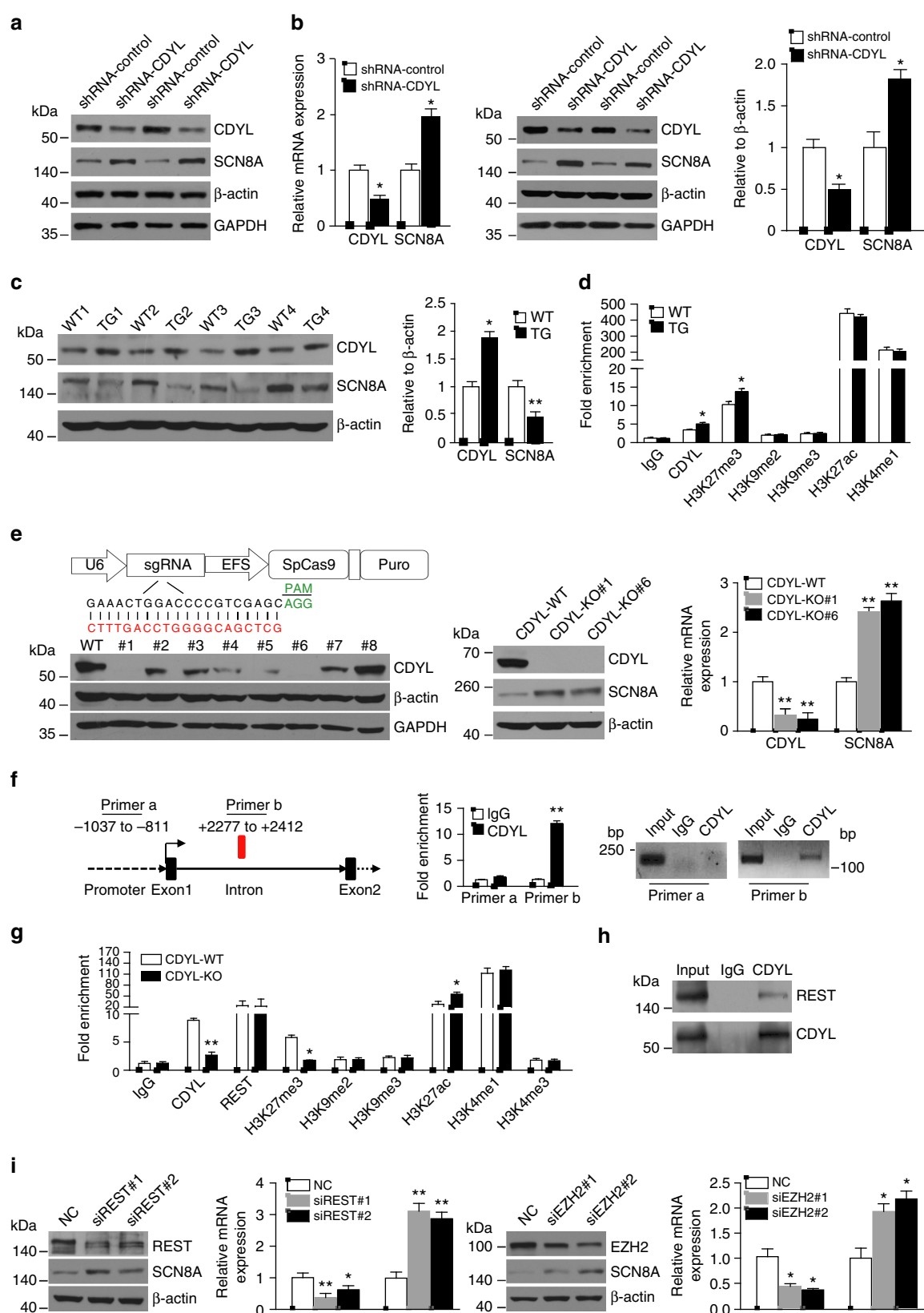

allowed measurement of neuronal AP threshold without inactivating voltage-gated ion channels. The results showed that infection of CDYL-shRNA lentiviruses significantly lowered the AP threshold, whereas other intrinsic membrane properties of neurons were not affected (Fig. 2d and Supplementary Table 1).

To further examine the function of CDYL on neuronal excitability in vivo, we generated a transgenic mouse model that over-expresses CDYL (Fig. 3a and Supplementary Fig. 4a–c). Intrinsic excitability of DG cells from transgenic mice over-expressing CDYL or the wild-type littermates were examined. Comparably, we found significantly elevated AP threshold, prolonged time delay to the first AP, and less AP firing in CDYL transgenic DG neurons (Fig. 3b–d), whereas other intrinsic membrane properties were not altered in these cells (Supplementary Table 2). Moreover, cortex and hippocampus CA1 neuronal cells from transgenic mice over-expressing CDYL also exhibited reduced intrinsic neuronal excitability (Supplementary Fig. 4d, e). Together, the above data indicated that CDYL inhibits intrinsic neuronal excitability in vivo.

**Genome-wide identification of chromatin targets of CDYL.** As CDYL was previously demonstrated to mainly function as a transcriptional corepressor[26–28], we next performed ChIP coupled with genomic sequencing (ChIP-seq) with mouse hippocampus to further decipher the molecular mechanisms of which CDYL inhibits intrinsic neuronal excitability. An antibody specific for mouse/rat CDYL was generated and used for this ChIP-seq assay. Total of 16,756 CDYL-specific binding peaks were identified using HiSeq2500 with a $p$ value cutoff of $10^{-3}$. Genomic distribution analysis indicated that the majority of CDYL binding sites (59.7%) were distant from proximal promoter regions. The second largest group of CDYL binding sites was located within gene bodies (exon and intron, 35.3%) and about 2.2% of the binding sites were located at promoter regions (Fig. 4a). Consistent with our previous ChIP-seq results obtained from human MCF-7 cells[28], recurrent appearance of the neuron-restrictive silencer element (NRSE, also known as RE1), the consensus REST binding sequence, was found in CDYL binding peaks (Fig. 4b). GO pathway analysis indicated that most CDYL target genes are involved in neurological functionality, of which genes involved in channel activity are most prominent (Fig. 4c).

To validate the ChIP-seq results at individual gene level, we performed quantitative ChIP (qChIP) analysis with mouse hippocampus on selected genes, including SCN8A, RIMS2,

GABRB3, CHRM4, GLRA1, CACNG3, NPAS4, and SCNN1G, representing each of the classified pathways of GO analysis. The enrichment of CDYL at these sites was confirmed (Fig. 4d). Significantly, BDNF, which we previously reported as a target gene of CDYL[29], was also identified in the current ChIP-seq analysis (Fig. 4d). Consistent with the function of CDYL as a transcriptional corepressor, infection of lentiviruses carrying CDYL-shRNA resulted in increased expression of the majority of CDYL target genes in mouse hippocampus (Fig. 4e). On the other hand, down-regulation of these genes was found in hippocampus from transgenic mice over-expressing CDYL (Fig. 4f).

Remarkably, SCN8A (the gene encodes voltage-gated sodium channel Nav1.6), a pivotal regulator of AP threshold and neuronal excitability[9], was identified as a target gene of CDYL by the ChIP-seq analysis. Quantitative real-time RT-PCR and western blot assays both confirmed that the level of SCN8A was increased in mouse/rat DG granule cells infected with CDYL-shRNA lentiviruses (Fig. 5a, b), whereas SCN8A expression was decreased in the hippocampus of transgenic mice over-expressing CDYL (Fig. 5c). We next performed ChIP analysis to investigate the molecular mechanism by which CDYL regulates SCN8A transcription. The CDYL binding peak resides within the intron region of SCN8A according to the ChIP-seq result, which was verified by the ChIP-qPCR assay performed with mouse hippocampal tissues (Fig. 5d). Strong enrichment of CDYL-associated H3K27me3, but not H3K9me2/3 activity was also found at the site (Fig. 5d). Significantly, specific enrichment of H3K27me3 was increased in the hippocampus of transgenic mice over-expressing CDYL (Fig. 5d).

To further examine how endogenous CDYL regulates SCN8A expression, we established a somatic CDYL-KO human SH-SY5Y cell line using CRISPR-Cas9 technology (Fig. 5e), as the amounts of CDYL-shRNA infected mouse hippocampal tissues were limited for ChIP assays. Western blot analysis and qRT-PCR measurements confirmed that CDYL expression was successfully eliminated in CDYL-KO cells, in which significant up-regulation of SCN8A was observed (Fig. 5e). We next performed ChIP analysis with soluble chromatin extracted from wild-type or CDYL-KO SY5Y cells using antibodies against CDYL, H3K9me3, H3K9me2, H3K27me3. The results showed that CDYL was strongly enriched in the intron, but not the promoter region of SCN8A in wild-type SY5Y cells (Fig. 5f). Significantly, knockout of CDYL resulted in decreased enrichment of H3K27me3, but not H3K9me2/3 (Fig. 5g), in line with the results obtained from

**Fig. 5** CDYL represses SCN8A transcription through an intronic element. **a**, **b** Knockdown of CDYL increased SCN8A expression. Hippocampal DG tissues were obtained from mice (**a**) or rats (**b**) infected with lentiviruses containing CDYL-shRNA or nonsilencing-shRNA for 14 days. Western blotting was used for mice experiments (**a**). For rats (**b**), the mRNA and protein levels of SCN8A and CDYL were examined by real-time RT PCR (*left*) and western blot analysis (*middle*) and quantified (*right*). $n = 3$, *$p < 0.05$, paired two-tailed Student's *t*-test. **c** SCN8A is downregulated in transgenic mice over-expressing CDYL. Hippocampus tissues were obtained from adult WT or CDYL-TG littermates. Western blot analysis and quantification were performed. $n = 4$, *$p < 0.05$, **$p < 0.01$, paired two-tailed Student's *t*-test. **d** Mouse hippocampus tissues were obtained from WT and CDYL-TG littermates. ChIP experiments were performed using the indicated antibodies. Real-time PCR assays were performed for the measurement. $n = 3$, *$p < 0.05$, paired two-tailed Student's *t*-test. **e** Somatic knockout of CDYL using CRISPR/Cas9 technology. Western blotting and RT-PCR assays were performed to confirm the elimination of CDYL in SH-SY5Y cells. Puromycin-resistant single cell-derived colonies were analyzed by western blotting and clone #1 and #6 were used for the subsequent functional studies. **$p < 0.01$, paired two-tailed Student's *t*-test. **f** CDYL binds to a regulatory element within SCN8A intron. qChIP assays were performed in SH-SY5Y cells with primer pairs specific to indicated regions (*left*). Normal rabbit IgG or antibody specific to CDYL was used to immunoprecipitate the protein-DNA complex (*middle*). Conventional semi-quantitative ChIP assays performed at the indicated regions (*right*). **$p < 0.01$, paired two-tailed Student's *t*-test. **g** Depletion of CDYL resulted in decreased regional enrichment of H3K27me3 and increased regional enrichment of H3K27ac. CDYL-KO SH-SY5Y cell lysates were collected, and ChIP-qPCR experiments were performed using the indicated antibodies. $n = 3$, *$p < 0.05$, **$p < 0.01$, paired two-tailed Student's *t*-test. **h** CDYL interacts with REST in vivo. Whole cell lysates from SH-SY5Y cells were prepared, and immunoprecipitated (IP) was performed with anti-CDYL followed by immunoblotted with antibodies against the indicated proteins. **i** Knockdown of REST or EZH2 in SY5Y cells increased SCN8A expression. The protein and mRNA levels were examined by Western blot analysis and real-time RT PCR. *$p < 0.05$, **$p < 0.01$, paired two-tailed Student's *t*-test. Data were represented as mean ± s.e.m.

CDYL transgenic hippocampus and supporting that CDYL recruits H3K27 meththyltransferase EZH2 to regulate target gene expression as we demonstrated in our previous work[29]. Significantly, ChIP results also showed strong enrichment of the sequence-specific transcription REST in the SCN8A intron region (Fig. 5g), consistent with the presence of the NRSE motif revealed by the ChIP-seq results and the in vivo interaction between REST

and CDYL (Fig. 5h)[28]. Note depletion of CDYL in cells did not affect the regional REST enrichment (Fig. 5g), supporting the role of REST as a sequence-specific transcription factor capable of direct interaction with NRSE. Knockdown of REST or EZH2 with specific siRNAs in SY5Y cells also lead to significant up-regulation of SCN8A (Fig. 5i), further supporting the role of REST and EZH2 in participating in SCN8A gene regulation.

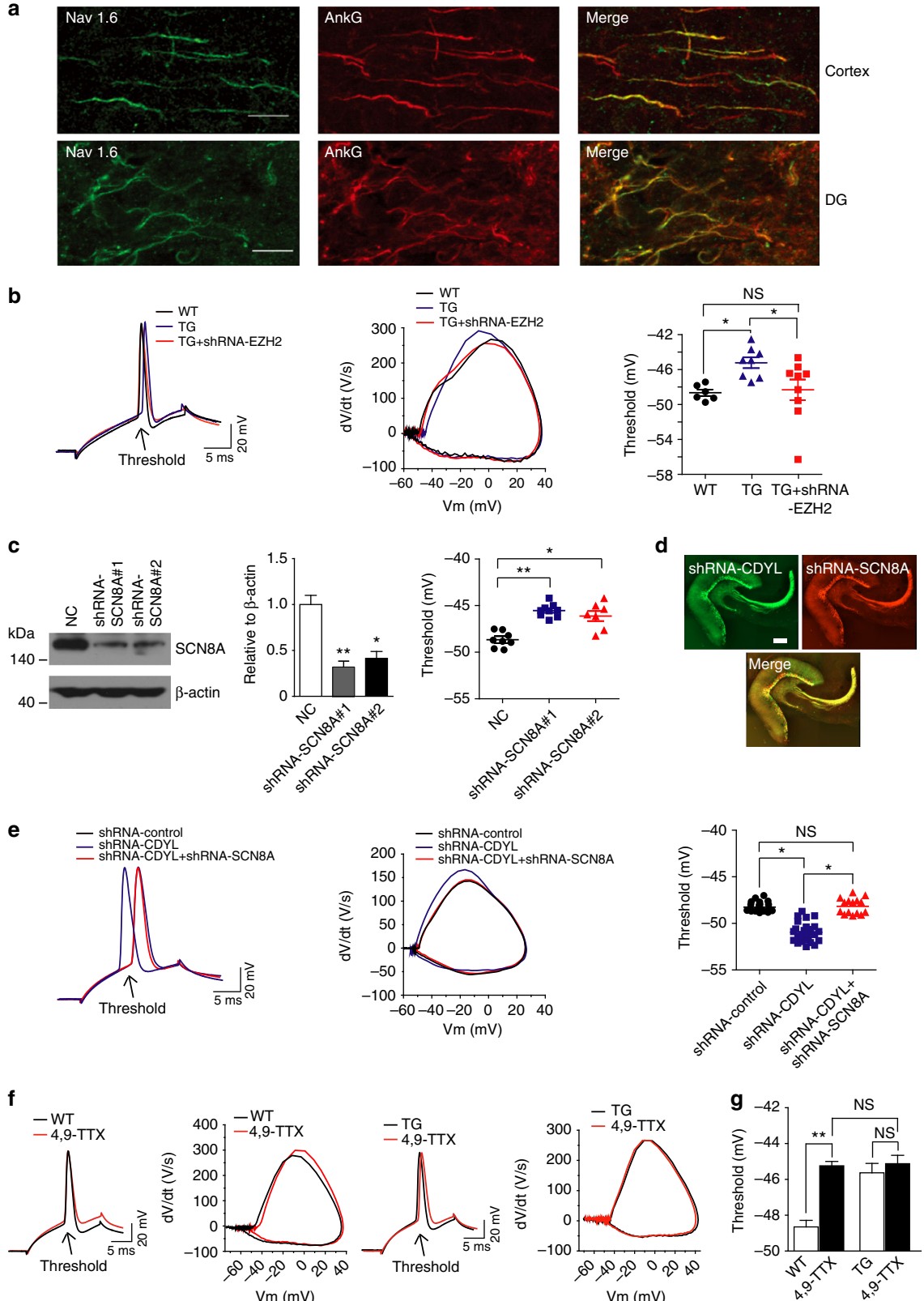

Interestingly, we found H3K4me1 and H3K27ac, two important marks of transcriptional enhancers, were also enriched at CDYL-present SCN8A intronic region (Fig. 5d, g). Knockout of CDYL in SY5Y cells resulted in increased enrichment of H3K27ac, whereas the level of H3K4me1 was not altered (Fig. 5g). The presence of both repressive H3K27me3 and active H3K4me1/H3K27ac marks indicated that this region resembles the bivalent chromatin domains found in embryonic stem cells, which play an essential role in regulation of key developmental genes[31]. Together, these results support an argument that CDYL inhibits SCN8A transcription mainly through recruiting H3K27me3 methyltransferase activity, possibly also the HDAC activity albeit to a less degree, to the intronic silencer of the gene.

**CDYL inhibits functional Nav1.6-mediated sodium currents.** Having established that CDYL inhibits SCN8A expression transcriptionally, we next examined whether and how CDYL regulates Nav1.6-mediated sodium currents and AP threshold in neuronal cells. We first examined the cellular distribution of Nav1.6 by immunostaining of mouse hippocampal DG granule cells. Consistent with previous studies in cortical neurons[9], the results showed that Nav1.6 was absent in the somatic membrane, but mainly localized to AIS of neurons, manifested by the co-localization of Nav1.6 with AnkG, a sodium channel-associated protein known to accumulate at the region (Fig. 6a). Further examination showed that the peak density of Nav1.6 staining was about 30–50 μm from the soma, corresponding to the AP initiation zone that was previously identified[32] (Fig. 6a).

To investigate whether CDYL regulates AP threshold by modulating the expression of Nav1.6, we first injected lentiviruses carrying EZH2-shRNA-GFP to the hippocampus of transgenic mice over expressing CDYL. Consistent with the above demonstrated role of EZH2 in coordinating with CDYL in regulating SCN8A expression (Fig. 5d, g, i), reducing EZH2 expression in the hippocampus of transgenic mice over-expressing CDYL alleviated CDYL-mediated changes of AP threshold (Fig. 6b), supporting the notion that CDYL coordinates with EZH2 to inhibit SCN8A expression and thus suppress intrinsic neuronal excitability.

To further confirm the altered SCN8A expression regulated by CDYL underlies the observed AP threshold plasticity, we sought to reduce the level of Nav1.6-mediated sodium channels in the mice injected with CDYL-shRNA to rescue the functional changes mediated by knocking-down CDYL. To this end, lentiviruses carrying mouse SCN8A-shRNA were generated and western blotting confirmed their effectiveness for silencing SCN8A expression (Fig. 6c). While injection of CDYL-shRNA-GFP into mouse DG area lead to reduced AP threshold compared to control group, stereotaxically co-injecting CDYL-shRNA-GFP and SCN8A-shRNA-mCherry completely abolished the AP threshold change caused by down-regulation of CDYL

(Fig. 6d, e), suggesting that CDYL-mediated functional changes in AP threshold are mainly mediated by SCN8A-encoded Nav1.6 sodium channels. Moreover, we obtained SCN8A null heterozygotes which exhibited reduced expression of SCN8A and higher AP threshold (Supplementary Fig. 5a, b)[33–35]. Injecting lentivirus carrying CDYL-shRNA into the DG cells of SCN8A null heterozygotes lead to increased expression of SCN8A and reversed the AP threshold to nearly as the same as the wild-type littermates (Supplementary Fig. 5a, b), further supporting CDYL affects AP threshold by inhibiting SCN8A expression.

Next, we treated CDYL transgenic or CDYL-shRNA infected mice with 4,9-TTX, a selective inhibitor of Nav1.6 channels[36, 37], to investigate whether functional Nav1.6 channels mediate CDYL's influence on AP threshold. The results showed that in the presence of 4,9-TTX, wild-type and CDYL transgenic neurons had identical AP threshold, indicating that blocking Nav1.6 channels abolished the effect of CDYL on AP generation (Fig. 6f, g). Similar results were also obtained from neurons infected with CDYL-shRNA, further supporting the notion that CDYL regulates AP threshold mainly through targeting functional Nav1.6 channels in granule cells (Supplementary Fig. 6).

Additionally, we investigated the role of CDYL in functional Nav1.6-mediated currents by performing voltage-clamp recordings to measure voltage-dependent sodium currents in rat hippocampal DG neurons. Neurons infected with CDYL-shRNA displayed significantly larger sodium currents density compared to the control cells (Fig. 7a, b), suggesting knockdown of CDYL lead to increased total sodium currents. We further analyzed the activation curves using the Boltzmann equation and calculated the half-activation voltage ($V_{1/2}$) and slope ($k$). CDYL-knockdown neurons have more negative half-activation voltage compared to the controls, with no significant difference between the fitting slopes (non-infected control: $V_{1/2} = -34.08 \pm 0.76$ mV, $k = 0.086 \pm 0.0037$; nonsilencing-shRNA: $V_{1/2} = -33.95 \pm 0.97$ mV, $k = 0.078 \pm 0.0069$; CDYL-shRNA: $V_{1/2} = -38.15 \pm 0.81$ mV, $k = 0.077 \pm 0.0035$) (Fig. 7c, d). This shift of activation curve could be because knockdown of CDYL leads to an increased proportion of Nav1.6-mediated currents in the total sodium currents, as Nav1.6 produces currents with more negative $V_{1/2}$ compared to other sodium channel subtypes in hippocampus (mainly Nav1.1 and Nav1.2-mediated currents)[38]. To test this hypothesis, we measured activation curves in the presence of 4,9-TTX. As expected, when blocking Nav1.6 channels, $V_{1/2}$ and peak sodium current obtained from CDYL-knockdown neurons or CDYL transgenic neurons were identical to that of control groups (Fig. 7e–h, and Supplementary Fig. 7), suggesting the shift of activation curve upon CDYL knockdown/over expression is mainly due to altered Nav1.6-mediated currents.

**CDYL suppresses seizure susceptibility.** Since SCN8A encoded Nav1.6 sodium channels play an important role in epileptic

**Fig. 6** Overexpression of CDYL in vivo inhibits functional Nav1.6 current. **a** Double staining for AnkG (*red*) and Nav1.6 (*green*) in cortex and hippocampal DG region of mice. *Scale bar*, 10 μm. The axon initial segment showed strong staining for Nav1.6. **b** Typical spikes (*left*), associated phase plane plots (*middle*), and the average spike threshold (*right*) obtained from DG neurons of WT, TG, and EZH2-shRNA-infected TG mice. *$p < 0.05$, one-way ANOVA with Bonferroni's multiple-comparison test. **c** Western blotting showing SCN8A was significantly reduced after SCN8A-shRNA lentivirus injection into mouse hippocampal DG area for 14 days (*left*), quantification was done by normalizing the level of SCN8A to that of β-actin (*middle*, $n = 3$), and the average spike threshold obtained from DG neurons of shRNA-Control (NC), shRNA-SCN8A#1 and shRNA-SCN8A#2, *$p < 0.05$, **$p < 0.01$, unpaired two-tailed Student's *t*-test. shRNA-SCN8A#1 was used for the subsequent experiments. **d** Expression of lentiviruses in 14 days after infecting the mouse hippocampal DG area. *Scale bar*, 200 μm. **e** Typical spikes (*left*), associated phase plane plots (*middle*), and the average spike threshold (*right*) obtained from DG neurons of shRNA-Control, shRNA-CDYL and both shRNA-CDYL and shRNA-SCN8A infected mice. *$p < 0.05$, one-way ANOVA with Bonferroni's multiple-comparison test. **f** Typical action potentials and associated phase plane plots obtained from DG neurons of WT and TG mice, in the presence and absence of 75 nM 4,9-TTX. **g** Graphs showing the average AP threshold before and after adding 4,9-TTX. **$p < 0.01$, NS means non-significant, one-way ANOVA with Bonferroni's multiple-comparison test. Data were represented as mean ± s.e.m.

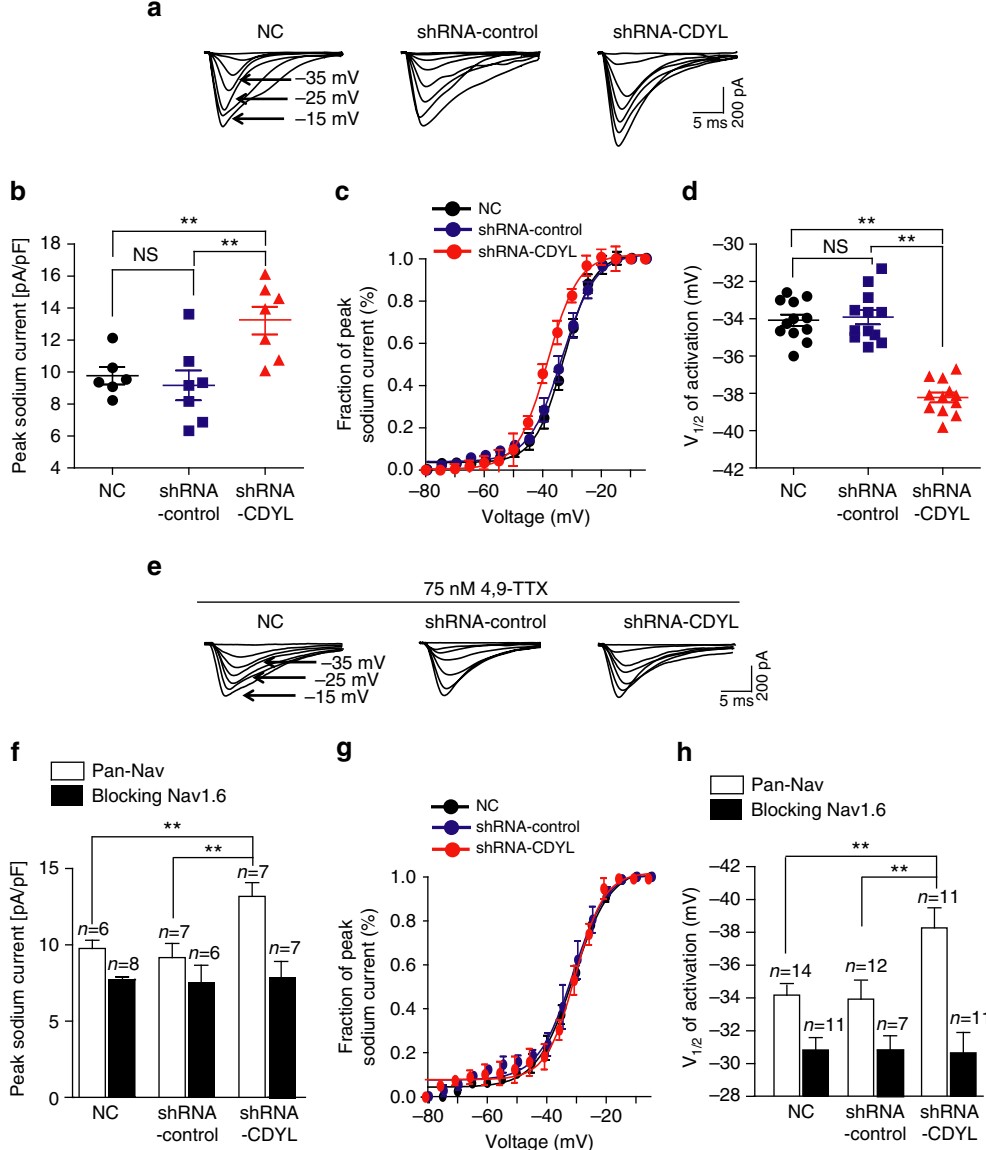

**Fig. 7** Knockdown of CDYL enhanced Nav1.6-mediated current. **a** Representitive sodium current traces obtained from normal control (NC), nonsilencing-shRNA-infected and CDYL-shRNA-infected rat DG neurons. Sodium currents were evoked by step depolarization from soma. **b–d** Graphs showing the density of peak sodium currents, activation curves and half-activation values of sodium currents, respectively. **p < 0.01, one-way ANOVA with Bonferroni's multiple-comparison test. **e** Representitive sodium current traces in the presence of 75 nM of 4,9-TTX. **f–h** Graphs showing the density of peak sodium currents, activation curves and half-activation values of sodium currents, respectively, in the presence of 75 nM of 4,9-TTX. The above experiments were all performed with rat DG neurons. **p < 0.01, one-way ANOVA with Bonferroni's multiple-comparison test. Data were represented as mean ± s.e.m.

encephalopathy[36, 39], to further assess the pathophysiological consequences of CDYL–SCN8A axis dysfunction in vivo, we next examined whether altered CDYL expression would affect seizure susceptibility in animal models. A commonly used KA model was used to assess epilepsy susceptibility, in which a single episode of Class V seizures (as defined by the Racine scale) or status epilepticus (SE) is induced in rodents by administration of KA, followed by termination with anticonvulsants such as sodium pentobarbital (SP) after 1 h. Spontaneously overt behavioral seizures commonly occur (defined as the onset of chronic TLE) in treated animals after a delay of a few weeks of latent period[40–42]. Depth electrodes were stereotaxically implanted in the DG of adults infected with lentiviruses carrying CDYL-shRNA or nonsilencing-shRNA. Administration of 10 mg kg⁻¹ KA intraperitoneally elicited SE in Control-shRNA infected rats in 107.2

± 4.89 min (n = 5, Fig. 8a, b), whereas the same treatment caused SE within 73 ± 3.91 min in CDYL-shRNA infected rats (n = 5, Fig. 8b). CDYL-shRNA infected rats exhibited significantly faster seizure progression, measured by maximum seizure class reached in each successive bin of 15 min (Fig. 8c). Moreover, the maximum seizure severity was considerably higher in CDYL-shRNA infected rats, with more of them developing into tonic hindlimb extension compared to controls (Fig. 8d). On the other hand, transgenic mice over-expressing CDYL had significantly slower seizure progression and reduced maximum seizure severity, and almost 80% of transgenic mice remained still without signs of seizure (n = 8, Fig. 8e, f). Clearly, these results indicate that altered CDYL expression in hippocampus has a substantial impact on the induction and development of acute epileptic seizures.

Next, we terminated the KA induced-SE with SP (30 mg kg⁻¹ imp.), and used electroencephalography (EEG) recordings and synchronous video monitoring to examine the latent period in 24-hour cycle of day and night over a period of 3 months. The "motor-seizure latent period duration" was calculated as the time taken for the onset of motor seizures[43], which was typically Class III (Supplementary Movie 1). In one case, we observed Class V seizure for onset of spontaneous epilepsy (Supplementary Movie 2). We observed that knockdown of CDYL resulted in overt motor convulsions within 10.9 days of halting SE (latent period duration=$10.8 \pm 1.39$ days, $n = 5$; Fig. 8g), whereas control rats displayed similar levels of spontaneous seizures 3 weeks later (latent period duration=$23.4 \pm 2.71$ days, $n = 5$; Fig. 8g). These results further support the notion that CDYL inhibits seizure susceptibility and latent period duration.

In order to investigate the role of Nav1.6 channels in CDYL-mediated change of seizure susceptibility, we examined the effect of simultaneously knocking-down CDYL and SCN8A on seizure susceptibility. Co-injecting lentiviruses carrying CDYL-shRNA-GFP and SCN8A-shRNA-mCherry reversed the behavioral alteration in seizure susceptibility induced by knock-down of CDYL (Fig. 8h, i), suggesting that CDYL regulates epilepsy

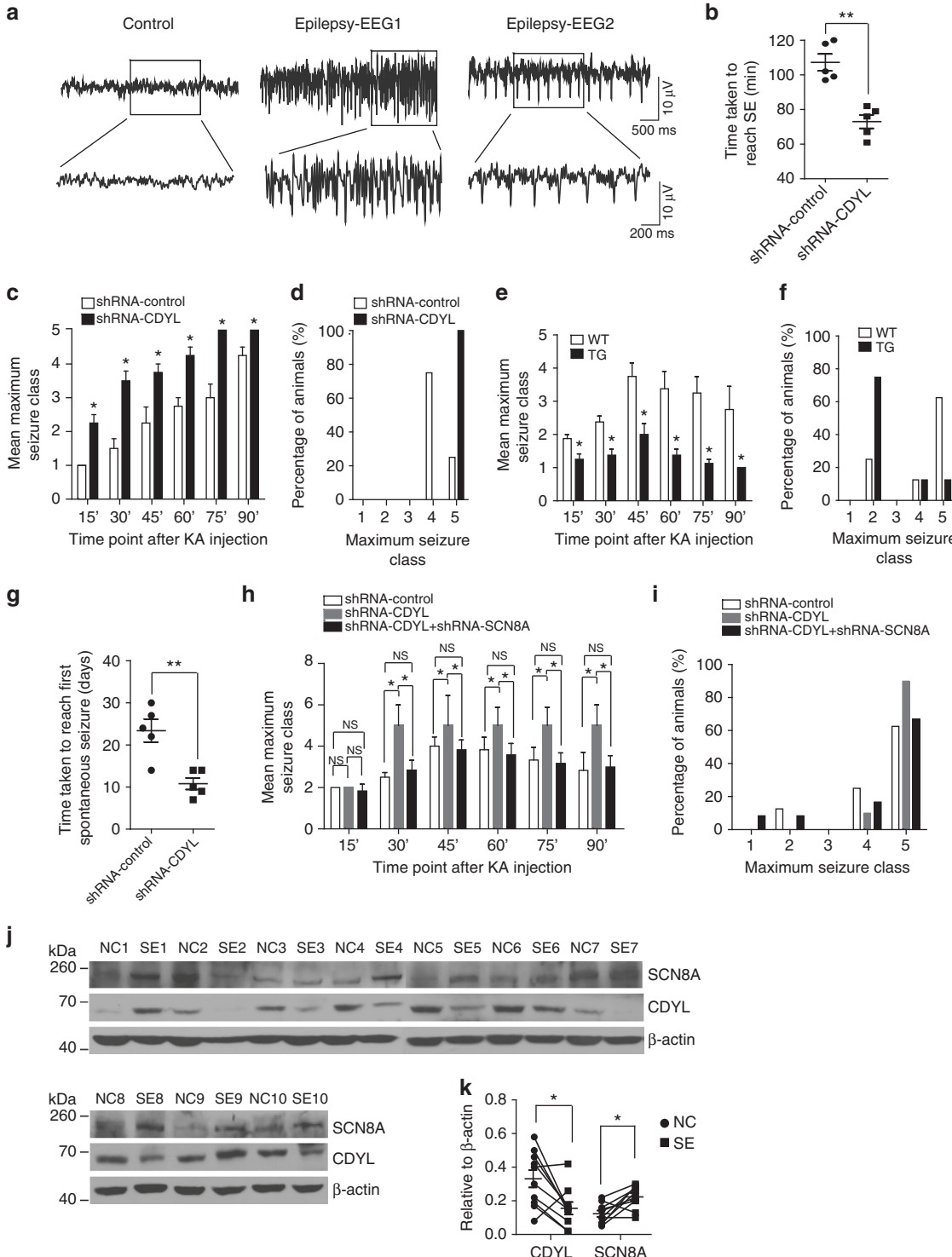

susceptibility mainly through Nav1.6 sodium channels. While *SCN8A* null heterozygotes exhibited less seizure susceptibility when compared to wild-type littermates, injecting lentiviruses carrying CDYL-shRNA to *SCN8A* null heterozygotes abolished the changes of epilepsy susceptibility in these mice (Supplementary Fig. 8a, b).

Finally, we examined the expression level of CDYL in hippocampus in surgically removed human epileptic tissues. Compared to adjacent normal tissues, 7 out of 10 matched pairs of TLE samples have decreased CDYL expression accompanied with increased SCN8A expression in the epileptic foci (Fig. 8j, k). Furthermore, immunostaining experiments confirmed the decrease in CDYL expression occurs in single neuronal level in epileptic hippocampus (Supplementary Fig. 8c, d). These results support the important role of CDYL–SCN8A axis in epileptogenesis.

## Discussion

In summary, we identified the transcriptional corepressor and chromodomain containing protein CDYL as a key regulator of activity-dependent intrinsic neuronal plasticity. CDYL undergoes protein degradation upon enhanced network activity, leading to reduced AP threshold and enhanced intrinsic excitability of neurons. We demonstrated that CDYL transcriptionally represses SCN8A, the gene encoding Nav1.6 sodium channels, such that the axonal Nav1.6 currents were reduced. Given that dysfunction of Nav1.6 currents is involved in altered learning, memory, and many neurological and psychiatric brain disorders, such as epilepsy, intellectual disability and pain[34, 36, 37, 44], our results suggest that modulating neuronal intrinsic excitability via the CDYL–SCN8A axis could play a key role in both physiological and pathological processes of the brain.

Hebbian plasticity is defined by synaptic-specific changes in strength driven by coordination of pre-synaptic input and post-synaptic depolarization. Numerous pieces of evidence have shown that induction and maintenance of Hebbian plasticity require activation of NMDA receptors (NMDAR), which allow $Ca^{2+}$ influx and subsequently activates signaling cascades that modify the strength of targeted synapses[25]. Interestingly, the classic methods used to induce Hebbian plasticity are also able to evoke non-Hebbian plasticity in different compartments of a neuron. For example, associative pairing of a highly synchronous synaptic input with APs leads to branch-specific dendritic plasticity through regulation of dendritic Kv4.2 potassium channels[45]. A classical procedure for Hebbian plasticity induction, theta-burst stimulation, can suppress global neuronal excitability in parallel with synaptic potentiation by regulating dendritic HCN channels[46]. Actually, in both cases modification of neuronal intrinsic plasticity is dependent on the activation of NMDAR. This phenomenon raises a possibility that opening of NMDAR could be a key messenger to send positive or negative feedback signals from synapses to a subdivision of a neuron. In our study, we demonstrated that activation of NMDAR resulted in reduced expression of CDYL (Fig. 1f), leading to enhanced neuronal intrinsic excitability (Figs. 2–3). These results suggest that CDYL could be an important mediator to link rapid synaptic modification (Hebbian plasticity) and long-lasting axonal plasticity (non-Hebbian plasticity). On the other hand, while we demonstrated CDYL inhibits intrinsic plasticity mainly through transcriptional repression of SCN8A, we have previously shown CDYL represses BDNF[29], which is critically involved in modulation of synaptic transmission[47]. Therefore, CDYL links Hebbian and non-Hebbian plasticity also through target gene regulation. Notably, although our current evidence clearly indicates CDYL–SCN8A axis is critical in epileptogenesis, we cannot exclude BDNF could also play a role during this process[48]. However, it was shown that conditional knockout of TrkB (the receptor for BDNF), but not BDNF, prevents epileptogenesis in the kindling model[49], suggesting regulation of BDNF expression is not likely the major contributor to the strong phenotype of transgenic mice over-expressing CDYL in resistance of seizure induction.

Accumulating evidence indicates that epigenetic factors are key regulators in normal brain development as well as various neurological disorders[19, 50]. In addition to the enzymes that directly add or remove covalent modifications on histones or DNA, it is more and more recognized that reader proteins also play an important role in neurological disorders. A well-known and extensively studied example is MeCP2, the methyl-CpG binding protein that is the causative factor of two severe and progressive neurological disorders, Rett syndrome, and MECP2 duplication syndrome, resulted from loss and gain of function, respectively[51]. Recently, Brd4, a member of the BET-domain containing protein family that binds to acetylated histones, was reported to activate transcription in neurons and BET inhibitor Jq1 blocks memory in mice[52]. Identifying the important function of the methyl-histone reader CDYL in the nervous system adds a new member to this family.

In addition to promoters, it is increasingly realized that remote enhancers or silencers play a key role in regulating gene transcription in the nervous system[20]. A recent genome-wide study in cortical neurons identified a subset of enhancers enriched for H3K4me1 and transcriptional coactivator CBP, and the increased H3K27ac after membrane depolarization functions to regulate activity-dependent transcription. The activation of these enhancers requires binding of transcription factor FOS, which was previously thought to bind primarily to promoters[53]. In our case, knockout of CDYL in SY5Y cells led to a significant decrease of

**Fig. 8** CDYL inhibits seizure susceptibility and latent period duration in temporal lobe epilepsy. **a** Examples of in vivo EEG recordings obtained from CDYL-shRNA-infected and nonsilencing-shRNA-infected rats. **b** Graph depicting the time taken to reach status epilepticus after kainic acid (KA) administration from CDYL-shRNA- infected and nonsilencing-shRNA-infected rats. $n = 5$, **$p < 0.01$, paired two-tailed Student's *t*-test. **c** Graph depicting the seizure progression in CDYL-shRNA-infected and nonsilencing- shRNA-infected rats, illustrated as mean maximum seizure class reached by 15, 30, 45, 60, 75, 90 min after KA administration. *$p < 0.05$, unpaired two-tailed Student's *t*-test. **d** Incidence of maximum seizure class reached during the course of the experiments in (**c**). **e** Graph depicting the seizure progression in WT and TG mice, illustrated as mean maximum seizure class reached by 15, 30, 45, 60, 75, 90 min after KA administration. *$p < 0.05$, paired two-tailed Student's *t*-test. **f** Incidence of maximum seizure class reached during the course of the experiments in (**e**). **g** Graph showing time taken to reach first spontaneous seizure after administration of KA from CDYL-shRNA-infected and nonsilencing-shRNA- infected rats. $n = 5$, **$p < 0.01$, paired two-tailed Student's *t*-test. **h** Graph depicting the seizure progression in shRNA-Control, shRNA-CDYL, and shRNA-CDYL plus shRNA-SCN8A infected mice, illustrated as mean maximum seizure class reached by 15, 30, 45, 60, 75, 90 min after KA administration. $n = 12$, *$p < 0.05$, one-way ANOVA with Bonferroni's multiple-comparison test. **i** Incidence of maximum seizure class reached during the course of the experiments in (**h**). **j** Western blot analysis of SCN8A and CDYL in the hippocampus from control individuals or individuals with TLE. $n = 10$. **k** Quantification of the results in H by normalizing the protein levels of SCN8A and CDYL to that of β-actin in control and TLE groups ($n = 10$). *$p < 0.05$, paired two-tailed Student's *t*-test. Data were represented as mean ± s.e.m.

H3K27me3 and a moderate increase of H3K27ac at SCN8A intron region, whereas over expression of CDYL in transgenic mice caused local increase of H3K27me3 but the level of H3K27ac was not obviously changed. These results probably reflect the dose-dependent effect in which complete ablation of CDYL in a cell line more dramatically affect regional histone modification as compared to mild overexpression of CDYL in experimental animals, and demonstrate that change of regional H3K27me3 is the major underlying mechanism for CDYL repression of SCN8A expression. Importantly, H3K4me1/ H3K27ac, which marks transcriptionally active chromatin, and H3K27me3, which marks transcriptionally repressive chromatin, are both enriched at this intronic silencer of SCN8A, suggesting it is an important bivalent element to regulate SCN8A expression. Interestingly, the transcriptional repressor REST was also found present at this site and knockdown of REST led to upregulation of SCN8A, indicating CDYL likely acts as a corepressor of REST in regulating SCN8A expression. While REST has previously been reported to recruit a number of epigenetic corepressors, including Sin3 complex, CoREST, G9a, LSD1, and CtBP to exert gene regulation function in the nervous system[54], the genome-wide interaction of CDYL and REST revealed by our ChIP-seq results and their specific function in regulating SCN8A expression, suggesting CDYL is another important corepressor of REST to exert function in regulating GE circuit excitability. Nevertheless, our study suggests that the CDYL–SCN8A axis is critical to regulate intrinsic plasticity of neuronal cells, thus may affect the occurrence and development of epilepsy.

## Methods

**Animals**. Male adult Sprague-Dawley rats and C57BL/6 mice (6–8 weeks) were purchased from Charles River Laboratories (Beijing). SCN8A null heterozygote mice were provided by Yousheng Shu (Beijing Normal University). Transgenic mice over-expressing CDYL were generated by Cyagen Biosciences. All animals were handled in strict accordance with the "Guide for the Care and Use of Laboratory Animals" and the "Principles for the Utilization and Care of Vertebrate Animals", and all animal work was approved by the Institutional Animal Care and Use Committee at Peking University. Each effort was made to minimize animal suffering and the number of animals used. The experiments were blind to viral treatment or drug treatment condition during behavioral testing.

**Antibodies and reagents**. Commercial antibodies used were: anti-FLAG (Sigma), anti-CDYL (Sigma, HPA035578), anti-REST (Santa Cruz, sc-25398), anti-EZH2 (Proteintech, 21800-1-AP), anti-HDAC1 (sc-7872), anti-HDAC2 (sc-7899), anti-SIRT1 (sc-15404), anti-FOS (ABclonal, A0236), anti-SET8 (CST, #2996), anti-UTX (sc-79334), anti-KDM5B (sc-67035), anti-CtBP (Proteintech, 10972-1-AP), anti-MeCP2 (ABclonal, A5694), anti-Nav1.6 (Alomone, ASC-009), anti-H3K9me2 (Abcam, ab1220), anti-H3K9me3 (Abcam, ab8898), anti-G9a (Millipore, 09-071), anti-H3K27me3 (Millipore, 07-449), anti-H3K27ac (Abcam, ab4729), anti-H3K4me1 (Abcam, ab8895), anti-H3K4me3 (Abcam, ab8580), anti-GAPDH and anti-Actin (MBL). A CDYL antibody that we generated (peptides antigen: DGFQGESPEKLDPVDQG and IDDRRDQPFDKRLRFSV; B&M) was used for ChIP-seq, western blotting with rat/mouse cell lysates, and immunohistochemistry. Protein A/G beads were from GE Healthcare Biosciences, protease inhibitor mixture cocktail was from Roche Applied Science. Tetrodotoxin, bicuculline, CGP 55845, and DL-AP5 were from Abcam. Neurobiotin was from Vector Laboratories. Streptavidin Alexa Fluor 488 was from Life Technologies. 4,9-TTX was from Focus Biomolecules. All other reagents were purchased from Sigma-Aldrich.

**Lentivirus infection and western blotting**. Lentiviruses carrying shRNA-targeting rat or mouse CDYL lentiviral vectors (GV118), were from Shanghai Gene Pharma. The viruses were used to infect C6 cells in the presence of Polybrene. Adult rat hippocampal DG granule cells were infected with proper lentiviruses containing non-silencing shRNA or CDYL shRNA. Seven days after transfection, neurons were collected and either RNA or protein was extracted for quantitative real-time RT-PCR or western blotting, respectively. Lentiviruses carrying shRNA-targeting mouse EZH2 (GFP vector) and SCN8A (Cherry vector) were from Shanghai Gene Pharma. The infection efficiency was confirmed by the expression of green fluorescent protein or cherry under microscopy. The CDYL shRNA sequence is: 5′-GGTACATCTCCATTCATGG-3′. The mouse EZH2 shRNA sequence is: 5′-GCAAATTCTCGGTGTCAAACA-3′. The mouse SCN8A#1 shRNA sequence is: 5′-TGCTCTTTGCCTTAATGAT-3′. The mouse SCN8A#2 shRNA sequence is: 5′-TGATGAAGTGAAACCTTTA-3′. The non-silencing shRNA sequence is: 5′-

GCAAATTCTCGGTGTCAAA-3′. For western blotting, membranes were incubated with appropriate antibodies for overnight at 4 °C followed by incubation with a secondary antibody. Immunoreactive bands were visualized using WESTERN Blotting Luminal Reagent (Santa Cruz Biotechnology) according to the manufacturer's recommendation. The bands were quantified by densitometry with ImageJ software. Uncropped scans of the most important blots and gels are shown in Supplementary Fig. 9.

**Cell culture and RNA interference**. Rat C6 and human SH-SY5Y cells were purchased from ATCC and maintained in DMEM supplemented with 10% fetal bovine serum (FBS). siRNAs were synthesized by Shanghai GenePharma Co., Ltd. The sequences are as follows: non-silencing siRNA sense: 5′-UUCUCCGAAC-GUGUCACGUt-3′; EZH2 siRNA#1 sense: 5′-CAUCGAAAGAGAAAUGGAAtt-3′; EZH2 siRNA#2 sense: 5′-CUAACCAUGUUUACAACUAtt-3′; REST siRNA#1 sense: 5′-CCAAACCCUUUCGCUGUAAtt-3′; REST siRNA#2 sense: 5′-GCAUCCUACUUGUCCUAAUtt-3′. siRNAs were transfected into SH-SY5Y cells with the final concentration of 30–40 nM using Lipofectamine RNAiMAX (Invitrogen) according to the manufacturer's instructions.

**Immunoprecipitation**. For co-immunoprecipitation experiments, SY5Y cells were lysed in lysis buffer (50 mM Tris-HCl, pH 7.4, 150 mM NaCl, 1 mM EDTA, 0.3% NP-40 and protease inhibitor mixture) for 20 min at 4 °C. This was followed by centrifugation at 13 000 g for 15 min at 4 °C. Protein supernatant was incubated with 2 μg CDYL antibody for 12 h at 4 °C with constant rotation; 50 μl of 50% protein A agarose beads was then added and the incubation was continued for an additional 2 h. Beads were then washed five times with the lysis buffer. Between washes, the beads were collected by centrifugation at 500 g for 3 min at 4 °C. The precipitated proteins were eluted from the beads by re-suspending the beads in 2×SDS-PAGE loading buffer and boiling for 10 min. The resultant materials from immunoprecipitation or cell lysates were then subjected to western blotting analysis.

**Sholl analysis**. Morphology of neurons was analyzed by Sholl analysis. Briefly, recorded neurons were filled with 0.4% neurobiotin. Then, the slices containing the fixed neurons cross-linked with 4% paraformaldehyde and stained with streptavidin Alexa Fluor 594 conjugate. Fluorescent images were obtained using a confocal microscope (PerkinElmer UltraVIEW-VoX) and imported into ImageJ (NIH) for further analysis. Concentric circles at 10 μm intervals were drawn around the soma[55]. The number of dendrites crossing each circle was calculated and the data were presented as mean ± s.e.m.

**Intracranial NMDA infusion**. Injection of NMDA (Sigma-Aldrich) were made with Hamilton syringes (Hamilton) that were connected to 30-gauge injectors (Plastics One, US). A volume of 1.0 μl of 0.5 μg μl⁻¹ NMDA[56] was infused into rat lateral ventricle (coordinates, bregma: anterior–posterior, −0.80 mm; dorsal–ventral, −1.60 mm; lateral, ±4.00 mm). NMDA was administered for 3 days before rat hippocampus was removed for western blotting and RT-PCR analysis.

**Enriched environment experiment**. For experiments in which rats were exposed to an enriched environment[57], sixteen SD rats (100–150 g) were divided into two groups: control and enriched environment. In the physically enriched condition, animals were housed in large three-floor cages for 4 weeks (±220 000 cm³; 72 cm × 45 cm × 69 cm) containing various inanimate attributes (tunnels, running wheel, extra nesting material) (four rats were kept in one cage, total n = 8). Controls were group-housed under standard conditions. All animals were housed with unlimited access to food and water in a day and night reversed cycle.

**DREADDs**. For experiments of DREADD, 10 weeks old CamkII-Cre mice (Jackson Lab, JAX-005359) were stereotaxically microinjected with rAAV-hSyn-DIO-hM3D (Gq)-mCherry bilaterally into dorsal hippocampus (coordinates, bregma: anterior–posterior, −1.50 mm; dorsal–ventral, −2.00 mm; lateral, ±1.00 mm, 500 nl per side). To minimize tissue injury, rAAV, or saline was slowly injected over 20 min using a pressure-injection system[58, 59]. Three weeks after rAAV delivery, mice received injections of vehicle or CNO (1.2 mg kg⁻¹) twice a day for 5 days. The mice were subsequently sacrificed and fluorescent tissue of hippocampus was collected under a fluorescent microscope (Olympus, Japan) for western blot analysis.

**Control tissues or tissues with TLE**. Patients (n = 355) with medically intractable TLE underwent phased presurgical assessment at Shengjing Hospital affiliated to China Medical University. Epilepsy diagnosis (including types and localization) was determined by clinical history, imaging examination (including MRI and/or PET), EEG (including scalp and/or intracranial EEG), and psychological assessment. Patients with TLE caused by stroke, tumor, injury, and malformations were excluded in this study. In those selected for surgery, the hippocampus was resected according to standard procedures. Between July, 2010, and February 2016, 246 hippocampi were resected. The study using clinical samples, which include 14 paired epileptogenic tissues and matched adjacent normal tissues, was approved by

the Ethics Committee of Shengjing Hospital affiliated to China Medical University (Supplementary Table 3). Tissues were frozen in liquid nitrogen immediately after surgical removal and maintained at −80 °C until mRNA and protein extraction. Informed consent was obtained from all subjects or their relatives.

**RT-PCR and real-time RT-PCR (qPCR).** Total RNA was isolated from samples with Trizol reagents (Invitrogen) and used for the first strand cDNA synthesis with the Reverse Transcription System (Roche). Any potential DNA contamination was removed by RNase-free DNase treatment (Promega). Relative quantitation was determined using the ABI PRISM 7500 sequence detection system (Applied Biosystems) that measures real-time SYBR green fluorescence and then calculated by means of the comparative Ct method ($2^{-\Delta\Delta Ct}$) with the expression of GAPDH as an internal control. The sequences of the primers used are provided in Supplementary Table 4.

**Chromatin immunoprecipitation and qChIP.** ChIP experiments were performed according to the procedure described previously[28]. Hippocampal tissues from 10 mice or about $1 \times 10^7$ SY5Y cells were fixed with 1% formaldehyde for 10 min at room temperature. The fixed cells were lysed in SDS lysis buffer (1% SDS, 5 mM EDTA and 50 mM Tris-HCl (pH 8.1), plus protease inhibitor cocktail). The lysates were then sonicated with $3 \times 10$ cycles (30 s on and off) (Bioruptor, Diagenode) to generate chromatin fragments of ~300 bp in length. Cell debris was removed by centrifugation and supernatant were collected. A dilution buffer (1% Triton X-100, 2 mM EDTA, 20 mM Tris-HCl (pH 8.1), 150 mM NaCl, plus protease inhibitor cocktail) was subsequently applied (1:10 ratio) and the resultant chromatin solution (aliquot 20 μl as the input) was then incubated with control or specific antibodies (3–5 μg) for 12 h at 4 °C with constant rotation. Protein A/G Sepharose beads (50 μl of 50% (vol/vol)) were added for incubation of another 2 h. Beads were collected by centrifugation at 500×g for 5 min at 4 °C. Beads were sequentially washed with the following buffers for 5 min at 4 °C: TSE I (1% Triton X-100, 0.1% SDS, 2 mM EDTA, 150 mM NaCl, 20 mM Tris-HCl (pH 8.1)); TSE II (1% Triton X-100, 0.1% SDS, 2 mM EDTA, 500 mM NaCl, 20 mM Tris-HCl (pH 8.1)); buffer III (1% Nonidet P-40, 0.25 M LiCl, 1% sodium deoxycholate, 1 mM EDTA, and 10 mM Tris-HCl (pH 8.1)); Tris-EDTA buffer. The input and the precipitated DNA–protein complex were decrosslinked at 65 °C for 12 h in elution buffer (1% SDS, 5 mM EDTA, 50 mM NaCl, 0.1 mg ml$^{-1}$ proteinase K, 20 mM Tris-HCl (pH 8.1)), and DNA was purified using PCR purification kit (Qiagen). Quantification of the precipitated DNA fragments were performed with real-time PCR using primers listed in Supplementary Table 5.

**ChIP sequencing.** A CDYL antibody that we generated (peptides as antigen: DGFQGESPEKLDPVDQG and IDDRRDQPFDKRLRFSV; B&M) was used for ChIP-seq. Note this antibody was also used for western blotting with rat/mouse cell lysates and immunohistochemistry. The specificity of this antibody was validated by peptide competition assay using the two CDYL antigenic peptides (Supplementary Fig. 2a), and its suitability for ChIP assay was confirmed by specific enrichment to known CDYL target such as BDNF promoter (Fig. 4d). The chromatin DNA precipitated by either normal rabbit IgG (control) or polyclonal antibody against mouse CDYL was purified with the Qiagen PCR purification kit. In-depth whole genome DNA sequencing was performed by the CapitalBio Corporation, Beijing. The raw sequencing image data were examined by the Illumina analysis pipeline, aligned to the unmasked mouse reference genome (GRCm38, mm10) using ELAND (Illumina), and further analyzed by MACS (model-based analysis for ChIP-seq). Enriched binding peaks of CDYL were generated after filtering through the control IgG. Genomic distribution of CDYL binding sites was analyzed by CEAS (the cis-regulatory element annotation system). The promoter region was defined as ≤+3 kb from the transcription start site, and "downstream" is defined as the distance from gene transcription stop site according to common criteria for ChIP-seq analysis[60]. De novo motif screening was performed on sequences ±125 bp from the centers of CDYL binding peaks using the CEAS and MEME systems. Pathway analysis was conducted based on Kyoto Encyclopedia of Genes and Genomes (Supplementary Table 6).

**Sterotaxically guided shCDYL injection and EEG recordings.** Animals (rats and mouse of 6–8 weeks) were deeply anaesthetized by intraperitoneally injection of SP (5 mg kg$^{-1}$ of body weight) and secured in the stereotaxic apparatus (RWD Ltd, China). 0.5 μl of Lentiviruses carrying CDYL-shRNA was bilaterally injected into hippocampal DG, CA1, CA3 regions: (rat DG: medial/lateral: +2.5 mm; anterior/posterior: −3.5 mm; dorsal/ventral: 2.5 mm below the dura. CA1: medial/lateral: +4.7 mm; anterior/posterior: −6 mm; dorsal/ventral: 3.5 mm below the dura. CA3: medial/lateral: +2.5 mm; anterior/posterior: −2.7 mm; dorsal/ventral: 2.5 mm below the dura. mouse DG: medial/lateral: +1.0 mm; anterior/posterior: −1.5 mm; dorsal/ventral: 2.0 mm below the dura). 10 min after microinjection, the needle was retracted and depth electrodes (Plastics One, USA) were implanted into the DG region. All animals were monitored for at least an hour post-surgery and at 12 h intervals for the next 5 days. Since CDYL-shRNA was strongly expressed from 7 DPI and lasted at least 3 months, the tested mice were used at 14 days DPI. EEG and synchronized video were recorded for 24 h a day with Omniplex-D Neural data acquisition system (Neurolog, USA) and an infrared camera, respectively.

**Generation of transgenic mice over-expressing CDYL.** Transgenic mice over-expressing CDYL were generated by Cyagen Biosciences. For transgenesis, ORF of mouse CDYL (1806 bp) with a FLAG tag was cloned into the mammalian expression vector pRP.Des2d. After digestion with NotI, linearized DNA was used for microinjection into the pronuclei of fertilized oocytes, derived from intercrosses of (C57BL/6×CBA) F1 mice. Transgenic mice were identified by PCR of tail-tip genomic DNA. The transgene PCR sense and antisense primers were 5′-GCTTTTGGAGTACGTCGTCT-3′ and 5′-GCAGGTGTTATCATCAGTAGGTT-3′, respectively. The internal control PCR sense and antisense primers were 5′-CAACCACTTACAAGAGACCCGTA-3′ and 5′-GAGCCCTTAGAAA-TAACGTTCACC-3′, respectively. Amplification of the transgene and internal control resulted in PCR products of 332 bp and 632 bp, respectively. Briefly, 2 μl of genomic DNA was amplified in 2×Premix Taq (TaKaRa), 0.8 μl of each transgene primer and 0.4 μl of each internal primer in a total volume of 25 μl. After an initial denaturation at 94 °C for 3 min, samples were amplified for 38 cycles (94 °C for 30 s, 57 °C for 35 s, 72 °C for 35 s). After the last cycle, samples were incubated for 5 min at 72 °C and resolved in 2% TAE-agarose gels. For the initial identification of transgenic mice over-expressing CDYL, out of 47 pups screened, 7 were identified positive (Supplementary Fig. 4a). Based on CDYL protein level examined by western blotting, we chose four of them as the founder mice to generate F1 (Supplementary Fig. 4b). Two founder mice successfully generated enough progeny, which expressed CDYL in the similar level and behaved similarly in the subsequent functional experiments we performed (Supplementary Fig. 4c).

**Acute slice preparation and electrophysiological recordings.** Horizontal slices containing hippocampus were obtained from 6–8 weeks old male Sprague Dawley rats, 6–8 weeks old transgenic mice over-expressing CDYL or the wild-type littermate controls, and C3Fe.Cg-Scn8a heterozygous mice (stock number 003798, Jackson Lab) or wide-type littermate controls[61]. In brief, animals were anesthetized and perfused intracardially with ice-cold modified "cutting solution" containing 110 mM choline chloride, 2.5 mM KCl, 0.5 mM CaCl$_2$, 7 mM MgCl$_2$, 25 mM NaHCO$_3$, 1.25 mM NaH$_2$PO$_4$, 10 mM glucose; bubbled continuously with 95%O$_2$/5%CO$_2$ to maintain PH at 7.2. The brain was then removed and submerged in ice-cold "cutting solution". Next, the brain was cut into 350 μm slices with a vibratome (WPI, USA). Slices were incubated in oxygenated (95% O$_2$ and 5% CO$_2$) "recording solution" containing: 125 mM NaCl, 2.5 mM KCl, 2 mM CaCl$_2$, 2 mM MgCl$_2$, 25 mM NaHCO$_3$, 1.25 mM NaH$_2$PO$_4$, 10 mM glucose (315 mOsm, PH 7.4, 37 ℃) for 30 min, and stored at room temperature.

Slices were subsequently transferred to a submerged chamber containing "recording solution" maintained at 34–36 °C. Whole-cell recordings were obtained from hippocampal DG, CA1, and cortical neurons. Pipettes had resistances of 5–8 MΩ. For current-clamp recordings, the external solution (unless otherwise noted) was supplemented with 0.05 mM APV, 0.01 mM CNQX, 0.01 mM bicuculline, and 0.001 mM CGP 55845 and internal pipette solution containing: 118 mM KMeSO$_4$, 15 mM KCl, 10 mM HEPES, 2 mM MgCl$_2$, 0.2 mM EGTA, and 4 mM Na$_2$ATP, 0.3 mM Tris-GTP, 14 mM Tris-phosphocreatinine (pH 7.3 with CsOH).

For morphological experiments, neurobiotin (0.2% w/v) was included in the intracellular pipette solution. Slices were fixed in 4% paraformaldehyde and stained with streptavidin Alexa Fluor 488 conjugate 24 h later[62]. For voltage-clamp recordings, the internal solution contained 140 mM CsMeSO$_4$, 10 mM HEPES, 2 mM MgCl$_2$, 1 mM EGTA, and 4 mM Na$_2$ATP, 0.3 mM Tris-GTP, 14 mM Tris-phosphocreatinin (pH 7.3, 295–300 mOsm). To isolate the Na$^+$ currents, CdCl$_2$ (200 mM), TEA (10 mM), and 4-AP (1 mM) were added. Families of Na$^+$ currents, in response to series of depolarizing voltage steps (30 ms) from a preceding prepulse of −100 (50 ms duration) to +20 mV, were recorded and the activation curves based on the peak currents at each step were generated. Activation threshold of the Na$^+$ currents was defined as the voltage at which the evoked peak current reached 10% of the maximum value.

Electrophysiological recordings were made using a Multiclamp 700B amplifier (Molecular Devices). Recordings were filtered at 10 kHz and sampled at 50 kHz. Data were acquired and analyzed using pClamp 10.0 (Molecular Devices). Series resistance was in the order of 10–30 MΩ and was approximately 60–80% compensated. Recordings were discarded if the series resistance increased by more than 20% during the course of the recordings.

**Kainic acid-induced status epilepticus.** KA (Sigma-Aldrich) was intraperitoneally administered to produce Class V seizures. The dose of kainite acid used was 10 mg kg$^{-1}$ for rats (6–8 weeks)[63] and 20 mg kg$^{-1}$ for mice (6–8 weeks)[49, 64]. To assess epilepsy susceptibility, seizures were rated using a modified Racine scale[65]: (1) immobility followed by facial clonus; (2) masticatory movements and head nodding; (3) continuous body tremor or wet-dog shakes; (4) unilateral or bilateral forelimb clonus; (5) rearing and falling. SE was terminated 1 h after onset with the use of SP (30 mg kg$^{-1}$; Sigma-Aldrich). Control groups were treated with SP only (30 mg kg$^{-1}$).

**Nav1.6 immunohistochemistry.** Mice were killed by perfusion with 1% paraformaldehyde and 1% sucrose (wt/vol) in 0.1 M phosphate buffer (PH 7.4) after deep anesthesia with SP. The brain was removed and post-fixed in the same fixative for 2 h, and subsequently immersed in 30% sucrose in 0.1 M phosphate buffer for

48 h. Cryostat coronal sections (20 μm) were obtained using a freezing microtome. The sections were rinsed in 0.01 M phosphate-buffered saline (PBS, pH 7.4) and incubated in a blocking solution (5% normal goat serum, 0.3% Triton X-100 in PBS, vol/vol) at 20–25 °C for 2 h, followed by overnight incubation at 4 °C with primary antibody to AnkG (1:200, Santa Cruz) and Nav1.6 (1:200, Alomone) in 0.1% Triton (vol/vol). After a complete wash in PBS, the sections were incubated in Alexa 488-conjugated goat anti-rabbit IgG and Alexa 594 goat anti-mouse IgG in 0.1% triton (1:1000; Molecular Probes) at 20–25 °C for 2 h. The sections were subsequently washed and mounted with Vecta shield mounting media (Vector Laboratories). Images were taken in the linear range of the photomultiplier with a laser scanning confocal microscope (ZEISS LSM 510 META NLO) and the projection of z stack images (0.4 μm per image) were used in the figures. For immunohistochemistry assay, tissues were prepared and subjected to immuno-histochemistry analysis with standard DAB staining kits (Gene Tech). The staining intensity was scored as follows: 0: negative; 1: weak; 2: moderate; 3: strong. The percent of positive cells was scored as follows: 0: negative; 1: 1–25%; 2: 26–50%; 3: 51–75%; 4: 76–100%.

**CRISPR-Cas9-mediated CDYL knockout**. A somatic CDYL-KO human SH-SY5Y cell line was established using CRISPR-Cas9 technology according to protocols previously described[66, 67]. The sequence 5′-GAAACTGGACCCCGTCGAGC-3′ was selected as the target sequence of CDYL-specific sgRNA, oligo 1: 5′-CACCGGAAACTGGACCCCGTCGAGC-3′; oligo 2: 5′-AAACGCTC-GACGGGGTCCAGTTTCC-3′. sgRNA was cloned into the pXPR_001 plasmid (AddGene plasmid 49535). pXPR_001 plasmids (with or without CDYL-sgRNA cloned) were next co-transfected into HEK293T cells with the packaging plasmids pVSVg (AddGene 8454) and psPAX2 (AddGene 12260) to produce lentivirus. SH-SY5Y cells were seeded in 6-well plates (at density ~1 × 10⁶ cells/well) before lentivirus infection. Puromycin-resistant single cell-derived colonies were analyzed by western blotting and DNA sequencing to confirm CDYL deletion.

**Statistical analysis**. For in vitro experiments, the cells were evenly suspended and then randomly distributed in each well tested. For in vivo experiments, the animals were distributed into various treatment groups randomly. Group data are represented as mean ± s.e.m. Comparisons between two groups were made using Student's paired or unpaired two-tailed $t$-test as appropriate. Comparisons among three or more groups were made using one-way ANOVA analyses followed by Bonferroni's multiple-comparisons test. No statistical methods were used to pre-determine sample sizes but our sample sizes are similar to those reported previously in the field[61, 62]. Data collection and analysis were not randomized or performed blind to the conditions of the experiments. Statistical significance of differences at $p < 0.05$ is indicated as asterisk (*), $p < 0.01$ is indicated as two asterisk (**) and $p < 0.001$ is shown with three asterisk (***) in all figures.

**Data availability**. CDYL ChIP-seq data has been deposited in GEO, and the accession number is GSE96721. All other data are presented in this manuscript and its supplementary files, or available from the authors on request.

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

## Acknowledgements

We thank Dr. Alasdair Gibb and Dr. Mala Shah for their valuable comments on this work. This work was supported by grants (973 Program: 2015CB559200 to Z.H., 2014CB542004 to J.L.) from the Ministry of Science and Technology of China, and grants (81572771 and 31371301 to J.L., 81371432 to Z.H., and 91219201 and 81130048 to Y.S.) from the National Natural Science Foundation of China.

## Author contributions

Y.L. performed and analyzed CRISPR-Cas9, ChIP-seq, ChIP-qPCR and western blotting. S.L. (Shirong Lai) performed and analyzed virus microinjection, KA-induced SE model, immunostaining of hippocampus, morphological experiments, EEG recordings and patch-clamp recordings. Y.L. and S.L. (Shumeng Liu) identified the transgenic mice over-expressing CDYL. Y.Z. performed bioinformatics of ChIP-seq. W.K. and Y.S. (Yousheng Shu) performed immunostaining of axonal Nav1.6 channels and provided SCN8A null heterozygotes. F.P. performed immunohistochemical staining of brain tissues. W.M. and S.L. (Shaoyi Li) provided human epileptic tissues and helped with western blot experiments and data analysis. C.Z. and M.Y. helped with CDYL virus injection and EEG recordings. Z.H. and J.L. designed the experiments, J.L., Z.H. and Y.S. (Yongfeng Shang) wrote the paper.

## Additional information

**Competing interests:** The authors declare no competing financial interests.

