## [Peer Review File · Nature Communications]

Reviewers' comments:

Reviewer #1 (Remarks to the Author):

In this interesting study, Liu et al. investigate the role of the epigenetic factor CDYL in modulating intrinsic neuronal excitability. Furthermore, they study whether this factor plays a role in epileptogenesis, using the kainate model of temporal lobe epilepsy (TLE). In addition, they report that CDYL expression is decreased in epileptic hippocampal foci of patients with epilepsy. Although the data are novel, several details are missing, and the description and interpretation of the data is not always precise, so that it is difficult to judge the impact of the reported findings. Examples follow.

(1) No details of the type of epilepsies (presumably partial epilepsies such as TLE) from which surgically resected tissues were obtained are given. Furthermore, "control" tissues are not defined (in the Methods section, "matched adjacent normal tissues" are mentioned). If CDYL is expressed in neurons, one would expect a decrease in CDYL expression in epileptic foci just because of the neuron loss in such tissue. It should be described exactly from which brain regions the resected tissues (epileptic and adjacent "control") were.

(2) Page 19: "The expression levels of SCN8A were mostly increased in these TLE samples": No, as shown in Fig. 8I, only half of the TLE samples were higher than control.

(3) It is reported that knockdown of CDYL reduced the duration of latent period after kainate-induced SE in rats from an average of 23 days to 10.9 days. However, it is not reported whether continuous (24/7) EEG/video recording was used to determine the onset of spontaneous seizures. Without such 24/7 recording it is not possible to correctly determine the latent period. In Fig. 8G, individual data should be shown for the latent period, because latent period typically varies in such models.

(4) How was "seizure" defined for determining latent period to onset of spontaneous seizures?

(5) It is often not clear which species (mice or rats) was used for a specific experiment.

(6) On p. 27, the dose of kainate is not described.

(7) Page 8: "Expression of CDYL ... was dramatically decreased". A decrease of ~40% (Fig. 1C) is not "dramatic".

(8) Page 23: "targeting CDYL may provide a robust method of dampening hyperexcitability that leads to seizures, opening new avenues of treatment for epilepsy patients" Really? An ideal anti-seizure drug should not dampen excitability. Which adverse effects would be associated with this strategy?

(9) Which behavioral alterations were observed in rats in which CDYL was inactivated? Similarly, how did the CDYL over-expressing mice behave?

(10) The review of Roopra et al. (Epilepsia 2012) on epigenetics and epilepsy should be cited and discussed. CDYL is mentioned in this review.

Reviewer #2 (Remarks to the Author):

This manuscript by Liu et al. investigates the role of Chromodomain Y-Like (CDYL) protein in intrinsic excitability and as an epigenetic regulator of the voltage-gated sodium channel gene, SCN8A. Overall, this is a clearly written manuscript and the findings are interesting. However, a number of issues need to be addressed.

Major concerns:

1. While the results of this study clearly demonstrate an effect of CDYL on SCN8A, it is less clear whether some of the described functional consequences (for example the effect of seizure susceptibility) are directly due to an effect on SCN8A. As pointed out by the authors, CDYL likely interacts with a large number of genes, including ion channels, and therefore observed functional consequences could be due to an effect on multiple genes. Definitive demonstration that altered

SCN8A expression underlies the observed functional changes would require deletion of the CDYL interaction site from the SCN8A intron. Since this experiment was not performed, the authors should discuss this caveat, and rewrite the appropriate sections of the manuscript to 'soften' the wording that attributes functional observations to a direct effect on SCN8A.

2. No reference was found for the 4,9-TTX compound. How did the authors establish the 4,9-TTX at 75mM is Nav1.6 selective?
3. Real time and Western data were normalized to GAPDH and beta-actin, respectively. However, since CDYL has many targets, was it confirmed that these control genes were not altered by CDYL manipulation?
4. Figure 4B: The authors state that the consensus motifs are similar for REST and CDYL, suggesting that they act as corepressors, though in Figure 5G there is no significant change in REST enrichment relative to input between the WT and CDYL-null cells. The authors should comment on this discrepancy. Additionally, there is inconclusive evidence that the two proteins functionally interact, unless the authors examine the relationship between the two proteins through binding assays.
5. It appears that the one transgenic founder was used to generate the mice for all experiments. Typically, at least two transgenic founders are used to eliminate possible effects due to the site of transgene insertion.
6. Figure 5D & 5G: While Figures 5D and 5G demonstrates that overexpression or knockdown of CDYL elicit significant changes in H3K27me3 at the SCN8A locus, there is an inconsistent effect on H3K27ac levels (no significant change in 5D). Authors should address this discrepancy.
7. The age of transgenic mice and shRNA rats tested for seizure susceptibility is unclear. Authors should also clarify how long after the shRNA injections the rats were tested.
8. Sample sizes were not provided for some experiments.
9. Were different clonal populations of the SH-SY5Y CDYL-KO cells tested in order to confirm that the phenotype was not due to nonspecific CRISPR background editing?

Minor concerns:

1. Source of the CamKII-Cre mice does not seem to be provided.
2. The dose of kainic acid administered was not provided.
3. Page 29, the reference given (Dudek et al. 2002) for the knockout of CDYL in SH-SY5Y cells seems incorrect.
4. Page 40, Figure 3 legend, 'mice's tails' is incorrect grammar.
5. Page 40, Figure 1 legend, 'hyperpolarizing' is spelt incorrectly.
6. Page 42, Figure 5 legend, does Part (A) and (B) correctly refer to mouse and rat data respectively?
7. Figure 4A: The authors should state their rationale for searching +3 kb of the promoter. It is

unclear how the authors defined "downstream" of the gene (polyA site, last coding exon, etc.).

8. Figure 4C: The authors should provide access to the other genes identified by GO analysis and ChIP-Seq against CDYL.

9. Figure 5C: TG4 does not appear to have knockdown of Nav1.6.

Reviewer #3 (Remarks to the Author):

Chromodomain on Y-like (CDYL) is a REST corepressor that physically bridges REST and the histone methylase G9a to repress transcription. Using a combination of molecular, electrophysiology and ChIP-seq, Liu et al show that whereas neuronal activity suppresses CDYL, CDYL suppresses neuronal excitability. The authors further identify the mechanism by which this occurs. Whereas shRNA to CDYL increases firing frequency and decreases the threshold for cell firing, overexpression of CDYL in transgenic mice reduces firing frequency and increases the threshold for cell firing. The authors further show that pharmacological inhibition of axonal Nav1.6 in WT DGCs mimics the effects of CDYL overexpression in WT cells (increases the threshold for cell firing, with little or no effect on cells overexpressing CDYL. The authors propose a model whereby CDYL puts a brake on neuronal excitability by tempering the Nav1.6-mediated sodium current. In support of the model, shRNA to CDYL elicits a marked increase in the Nav1.6-mediated sodium current and 4,9-TTX, an inhibitor of Nav1.6, rescues the elevated sodium current observed in cells expressing CDYL shRNA. Using a kainic acid model of status epilepticus, the authors show that whereas shRNA to CDYL markedly increases seizure progression and severity, overexpression of CDYL slows them. Finally, the authors show that CDYL directly binds the SCN8A promoter. Moreover, in cells overexpressing CDYL, H3K27me3 (a mark of gene repression) is enriched, with little or no change in H3K27ac (a mark of active gene transcription). In cells lacking CDYL, H3K27me3 is decreased, and H3K27ac is modestly increased. The authors conclude that CDYL-SCN8A axis affords a novel mechanism to regulate epileptogenesis and the neuronal hyperexcitability that induces seizures.

The present paper builds on a previous paper by several of the same authors that shows that CDYL is a negative regulator of dendritic arborization. The present paper is novel in that it is the first to address a role for the epigenetic factor CDYL in neuronal excitability and epileptogenesis. The experiments appear, for the most part, to be carefully executed. These positive aspects notwithstanding, there are several methodological and presentational aspects that should be addressed:

Major points:

1. Fig. 1: the quality of the representative Westerns shown in panels 1B, 5A and 5C are suboptimal and should be replaced.
2. Fig. 1D-H: The authors show that three diverse stimuli known to increase network excitability, enriched environment, injection of NMDA or kainic acid, all decrease CDYL expression in the hippocampus. Many studies have shown that enriched environment reduces spontaneous seizures and neuronal death, and improve cognition in the face of epileptogenic seizures. Yet the authors report that overexpression of CDYL reduces seizure progression and severity. Given that enriched environment decreases CDYL expression, how do the authors interpret this apparent discrepancy?
3. Fig. 5: Authors show that SY5Y cells lacking CDYL exhibit a marked decrease in H3K27me3 and increase in H3K27ac enrichment at the SCN8A promoter. They conclude that CDYL inhibits SCN8A transcription by recruiting H3K27 methyltransferase and the epigenetic remodeling enzymes HDAC1 and 2 to an intronic region of the gene and induce gene silencing. However, CDYL

overexpression does not decrease H3K27ac enrichment. The authors should examine a potential causal relation between CDYL and histone modifications. Moreover, the authors cannot rule out the possibility that REST, a master repressor of neural genes that silences gene transcription via histone acetylation, or EZH2, a repressor that confers a trimethylation mark at H3K27, act as corepressors to silence SCN8A. In light of findings by other that CDYL is a REST corepressor that physically bridges REST and the histone methyltransferase G9a (Mulligan et al, 2008 Mol Cell), and that CDYL interacts with EZH2 directly to promote trimethylation of H3K27 at the promoters of target genes such as BDNF (Qi et al, J Neurosci, 2014), the author should perform additional experiments to examine a role for REST and EZH2 in repression of SCN8A.

3. p22-23: Authors conclude that, "the CDYL-SCN8A axis is critical to regulate intrinsic plasticity of neuronal cells and targeting CDYL may provide a robust method of dampening hyperexcitability that leads to seizures." The paper would be substantially strengthened were the authors to examine a potential role for REST and/or EZH2 in these actions.

4. The findings demonstrating the impact of shRNA-CDYL on Nav1.6 activity would be further strengthened were the authors to determine whether overexpression of Nav1.6 could recapitulate the impact of CDYL knockdown on neuronal excitability.

Point-to-point response to reviewers' comments

Reviewer #1 (Remarks to the Author):

In this interesting study, Liu et al. investigate the role of the epigenetic factor CDYL in modulating intrinsic neuronal excitability. Furthermore, they study whether this factor plays a role in epileptogenesis, using the kainate model of temporal lobe epilepsy (TLE). In addition, they report that CDYL expression is decreased in epileptic hippocampal foci of patients with epilepsy. Although the data are novel, several details are missing, and the description and interpretation of the data is not always precise, so that it is difficult to judge the impact of the reported findings. Examples follow.

(1) No details of the type of epilepsies (presumably partial epilepsies such as TLE) from which surgically resected tissues were obtained are given. Furthermore, “control” tissues are not defined (in the Methods section, “matched adjacent normal tissues” are mentioned). If CDYL is expressed in neurons, one would expect a decrease in CDYL expression in epileptic foci just because of the neuron loss in such tissue. It should be described exactly from which brain regions the resected tissues (epileptic and adjacent “control”) were.

Authors: We thank the reviewer's comments and we have clarified these important details in the revised manuscript. In our study, all surgically resected human tissues were from 14 patients with medically intractable mesial temporal lobe epilepsy. Before surgical resection of hippocampi, the multi-channels intracranial EEG electrodes were implanted to localize the origin of epilepsy. The “epileptic tissues” in our study were from originating sites of spontaneous seizure which were determined

by their electrical seizure-like activities recorded with intracranial EEG electrodes. Tissues adjacent to “epileptic tissues” serve as the “controls”, which are normally the adjacent entorhinal cortical area. The control tissues could also show abnormal electrical activities during a seizure, but the occurrence of such activities is significantly later than that in epileptic tissues and we thus consider it as secondary effect. The detailed screening procedures for patient diagnosis and tissue selection have been clarified in “Control tissues and tissues with TLE” section of supplementary experimental procedures in the revised manuscript.

We provided two lines of evidence to exclude that neuron loss is the major reason for the decreased expression of CDYL in epileptic foci. We normalized the loading of total protein lysates by internal control β -actin from “control” and “epileptic tissues” (Fig. 8J-K) in the western blot analysis. Moreover, we performed immunostaining experiments with samples resected from epileptic human tissues. The results demonstrated the CDYL staining was significantly decreased in epileptic hippocampus in single neuron level in TLE patients (Supplementary Fig. 8C-D).

(2) *Page 19: “The expression levels of SCN8A were mostly increased in these TLE samples”: No, as shown in Fig. 8I, only half of the TLE samples were higher than control.*

Authors: In Fig. 8J (Fig. 8I in the original manuscript), western blotting showed that seven out of ten matched pairs of TLE samples have decreased CDYL expression accompanied with increased SCN8A expression (#3, #4, #5, #6, #7, #8, #10) (Fig. 8K). We have reworded the relevant sentences in the revised manuscript to describe these results more accurately.

(3) *It is reported that knockdown of CDYL reduced the duration of latent period after kainate-induced SE in rats from an average of 23 days to 10.9 days. However, it is not reported whether continuous (24/7) EEG/video recording was used to determine the onset of spontaneous seizures. Without such 24/7 recording it is not possible to correctly determine the latent period. In Fig. 8G, individual data should be shown for the latent period, because latent period typically varies in such models.*

Authors: We thank the reviewer for raising this point. We did use 24/7 EEG/video recording in these experiments and the relevant protocol was described in revised manuscript (line 16 to 19 in page19). Individual data for latent period have been shown in Fig.8G in the revised manuscript.

(4) *How was “seizure” defined for determining latent period to onset of spontaneous seizures?*

Authors: Here, we used “motor-seizure latent period” as previously described in Williams et al., 2009 to determine latent period, which is the time from the day of kainate treatment to the first convulsive motor seizure. The first seizure was confirmed by both EEG and video recordings. The onset of spontaneous seizure, as defined by modified Racine scale (See method section), is typically Class III seizure (Supplementary Movie 1), and in one case Class V seizure was observed (Supplementary Movie 2). We describe this at line 19 to 22 in page19.

Williams PA, White AM, Clark S, Ferraro DJ, Swiercz W, Staley KJ, Dudek FE. Development of spontaneous recurrent seizures after kainate-induced status epilepticus. *J Neurosci* 2009; 29:2103–2112.

(5) *It is often not clear which species (mice or rats) was used for a specific experiment.*

Authors: We have clarified the species (mice or rats) used in each experiment in methods and the relevant figure legend in the revised manuscript.

(6) *On p. 27, the dose of kainate is not described.*

Authors: The dose of kainate acid used was 10mg/kg for rats (Shah et al. 2004) and 20mg/kg for mice (He et al. 2004, Huang et al. 2009). We have clarified this in the “EXPERIMENTAL PROCEDURES” section of revised manuscript (Kainic Acid-induced Status Epilepticus).

He XP, Kotloski R, Nef S, Luikart BW, Parada LF, McNamara JO. Conditional deletion of TrkB but not BDNF prevents epileptogenesis in the kindling model. *Neuron* 2004; 43:31-42.

Shah MM1, Anderson AE, Leung V, Lin X, Johnston D. Seizure-induced plasticity of h channels in entorhinal cortical layer III pyramidal neurons. *Neuron* 2004; 44(3):495-508.

Huang Z, Walker MC, Shah MM. Loss of dendritic HCN1 subunits enhances cortical excitability and epileptogenesis. *J Neurosci.* 2009; 29(35):10979-88.

(7) *Page 8: “Expression of CDYL ... was dramatically decreased”. A decrease of ~40% (Fig. 1C) is not “dramatic”.*

Authors: Sorry for the misunderstanding. According to the quantification of the western blot, the decreased expression of CDYL in mice exhibited enhanced network activities was statistically significant (Fig. 1C). Therefore, in the revised manuscript we changed the word “dramatically” (line 1 in page 8) to “significantly” for more

accurate description.

(8) Page 23: “targeting CDYL may provide a robust method of dampening hyperexcitability that leads to seizures, opening new avenues of treatment for epilepsy patients” Really? An ideal anti-seizure drug should not dampen excitability. Which adverse effects would be associated with this strategy?

Authors: We thank the reviewer for raising this point. In our experiments, we found a decreased expression of CDYL in epileptic tissues from animal model and human patients. Moreover, over-expression of CDYL reduced the seizure susceptibility and knocking-down CDYL increased the seizure susceptibility. Therefore, we proposed that targeting CDYL may affect the occurrence and development of epilepsy. As the reviewer pointed out, simply dampen neuronal excitability could be associated with adverse effects such as cognitive deficit and impairment in learning and memory. Whether targeting CDYL would lead to these kinds of behavior changes needs further investigation. We have rewritten this part of discussion for more accurate description “Nevertheless, our study suggests that the CDYL-SCN8A axis is critical to regulate intrinsic plasticity of neuronal cells, thus may affect the occurrence and development of epilepsy” (line 6 to 8 in page 25).

(9) Which behavioral alterations were observed in rats in which CDYL was inactivated? Similarly, how did the CDYL over-expressing mice behave?

Authors: This is a very interesting question. Preliminary experiments conducted with mice over-expressing CDYL or rats down-regulating CDYL did not show obvious behavior alterations. Since several brain regions are involved in physiological conditions such as cognition and emotion and pathological conditions such as depression, we examined the expression of CDYL in the relevant brain areas in these animal models and found that alteration in CDYL expression was in a brain region-dependent manner. We are currently performing more experiments to further investigate the functional role of CDYL in different physiological and pathological brain processes.

(10) The review of Roopra et al. (*Epilepsia* 2012) on epigenetics and epilepsy should be cited and discussed. CDYL is mentioned in this review.

Authors: We thank the reviewer’s suggestion and we have cited and discussed this paper in our revision (line 1 to 3 in page 25).

Roopra A et al, Epigenetics and Epilepsy. *Epilepsia* 2012; 53(Suppl 9): 2–10.

Mulligan P et al, CDYL bridges REST and histone methyltransferases for gene repression and suppression of cellular transformation. *Mol Cell*. 2008; 32(5):718-26.

Reviewer #2 (Remarks to the Author):

This manuscript by Liu et al. investigates the role of Chromodomain Y-Like (CDYL) protein in intrinsic excitability and as an epigenetic regulator of the voltage-gated sodium channel gene, SCN8A. Overall, this is a clearly written manuscript and the findings are interesting. However, a number of issues need to be addressed.

Major concerns:

1. While the results of this study clearly demonstrate an effect of CDYL on SCN8A, it is less clear whether some of the described functional consequences (for example the effect of seizure susceptibility) are directly due to an effect on SCN8A. As pointed out by the authors, CDYL likely interacts with a large number of genes, including ion channels, and therefore observed functional consequences could be due to an effect on multiple genes. Definitive demonstration that altered SCN8A expression underlies the observed functional changes would require deletion of the CDYL interaction site from the SCN8A intron. Since this experiment was not performed, the authors should discuss this caveat, and rewrite the appropriate sections of the manuscript to 'soften' the wording that attributes functional observations to a direct effect on SCN8A.

Authors: We thank the reviewer's constructive comments. To solidify that modulating SCN8A expression underlies the effect of CDYL on neuronal intrinsic excitability and seizure susceptibility, we further performed two types of rescuing experiments in animal models. First, we generated lentiviruses carrying shRNA-SCN8A. While injection of lentivirus carrying shRNA-CDYL in rats significantly lower neuronal AP threshold and facilitate the development of epilepsy (Fig. 2D and Fig. 8B-D), coinjection of shRNA-SCN8A with shRNA-CDYL to the mice hippocampal DG neurons completely abolished the AP threshold changes (Fig. 6D-E) and reversed the alteration in seizure susceptibility (Fig. 8H-I). In addition, we obtained SCN8A null heterozygote mice from our collaborator Prof. Yousheng Shu (Li et al., 2014). These mice have reduced expression of SCN8A protein (Supplementary Fig. 5A) consistent with previously reports (Papale et al., 2009; Yin et al., 2015) and consequently exhibited more depolarized neuronal threshold (Supplementary Fig. 5B) and lower epilepsy susceptibility (Supplementary Fig. 8A-B) when compared with control mice. We demonstrated that SCN8A null heterozygotes injected with lentiviruses carrying CDYL-shRNA showed increased expression of SCN8A, recovered AP threshold (Supplementary Fig. 5A-B), and partially reversed alteration in seizure susceptibility (Supplementary Fig. 8A-B). These data support SCN8A-encoded Nav1.6 sodium channels mainly underlie the CDYL-mediated functional changes on AP threshold and epilepsy susceptibility.

Li T et al, Action potential initiation in neocortical inhibitory interneurons. *PLoS Biol.* 9; 12(9): e1001944 (2014).

Papale et al, Heterozygous mutations of the voltage-gated sodium channel SCN8A are associated with spike-wave discharges and absence epilepsy in mice. *Hum Mol Genet.* 18(9): 1633-41, (2009).

Yin L et al, Selective Modulation of Axonal Sodium Channel Subtypes by 5-HT1A Receptor in Cortical Pyramidal Neuron. *Cereb Cortex.* pii: bhv245 (2015).

2. *No reference was found for the 4,9-TTX compound. How did the authors establish the 4,9-TTX at 75mM is Nav1.6 selective?*

Authors: 4,9-TTX is a potent, selective sodium channel Na(v1.6) blocker (IC₅₀ = 7.8 nM), which exhibits greater potency for Nav1.6 over other TTX-sensitive subunits (IC₅₀ values are 1.26 μM (Nav1.2), 341 nM (Nav1.3), 988 nM (Nav1.4), 78.5 μM (Nav1.5) and 1.27 μM (Nav1.7)) (Rosker C et al., 2007; Hargus NJ et al., 2013). Here, we used 75 nM of 4,9-TTX, which is the selective dose for Nav1.6 sodium channels (Rosker C 2007; Hargus NJ et al., 2013). We have added the reference in the revised manuscript (Reference No. 40 and 41, Line 1 in Page 17).

Hargus NJ et al, Evidence for a role of Nav1.6 in facilitating increases in neuronal hyperexcitability during epileptogenesis. *J Neurophysiol* 110:1144-57 (2013).

Rosker C et al, The TTX metabolite 4,9-anhydro-TTX is a highly specific blocker of the Na(v1.6) voltage-dependent sodium channel. *Am J Physiol Cell Physiol* 293:C783-9 (2007).

3. *Real time and Western data were normalized to GAPDH and beta-actin, respectively. However, since CDYL has many targets, was it confirmed that these control genes were not altered by CDYL manipulation?*

Authors: Change of CDYL expression does not affect the expression of GAPDH and β-actin as we confirmed by western blotting. We provided the data in Fig.5A, Fig.5B and Fig. 5E in the revised manuscript. In addition, GAPDH and beta-actin were not identified as CDYL targets in our ChIP-seq data, which has been deposited in GEO with the accession number GSE96721.

4. *Figure 4B: The authors state that the consensus motifs are similar for REST and CDYL, suggesting that they act as corepressors, though in Figure 5G there is no significant change in REST enrichment relative to input between the WT and CDYL-null cells. The authors should comment on this discrepancy. Additionally, there is inconclusive evidence that the two proteins functionally interact, unless the authors examine the relationship between the two proteins through binding assays.*

Authors: We thank the reviewer for raising this point. The connection between

CDYL and REST was initially proposed by Dr. Yang Shi's group in an earlier study (Mulligan et.al, 2008), in which they performed GST pull-down assays to show CDYL directly interact with REST. We confirmed the interaction between CDYL and REST in neuronal SH-SY5Y cells by co-IP assays (Fig. 5H). In our current ChIP-seq performed in mouse hippocampus, we found recurrent appearance of REST motif in CDYL binding peaks, supporting the genome-wide functional connection between REST and CDYL. Since REST is a sequence-specific transcription factor containing a DNA binding domain, whereas CDYL does not have a DNA-binding domain and is thus a transcriptional corepressor likely recruited by REST, knockdown of CDYL should not affect the binding between REST and its target DNA sequence. We have explained this in the result section relevant to Fig. 4B in the revised manuscript.

Mulligan et al, CDYL bridges REST and histone methyltransferases for gene repression and suppression of cellular transformation. *Mol Cell*. 2008; 32:718-726

5. It appears that the one transgenic founder was used to generate the mice for all experiments. Typically, at least two transgenic founders are used to eliminate possible effects due to the site of transgene insertion.

Authors: We thank the reviewer for the comments and we have clarified this in the revision. Transgenic mice over-expressing CDYL were generated by microinjection of linearized DNA containing ORF of mouse CDYL (1806bp) with a FLAG tag cloned into the mammalian expression vector pRP.Des2d into the pronuclei of fertilized oocytes. For the initial identification of transgenic mice over-expressing CDYL, out of 47 pups screened, 7 were identified positive. According to CDYL expression level assessed by western blotting, we chose 8, 24, 31, 40 as the founder mice to generate F₁. Eventually, founder mice 8 and 24 generate enough progeny for subsequent experiments. The expression level of CDYL in these mice is similar and the mice from both founders behave similarly in the experiments we performed (Supplementary Fig. 4A-C). The relevant description of the method has been provided in "Generation of Transgenic Mice Over-expressing CDYL" section.

6. Figure 5D & 5G: While Figures 5D and 5G demonstrates that overexpression or knockdown of CDYL elicit significant changes in H3K27me3 at the SCN8A locus, there is an inconsistent effect on H3K27ac levels (no significant change in 5D). Authors should address this discrepancy.

Authors: Following the reviewer's suggestion, we have discussed this and performed additional experiments to address this issue. In Fig. 5D, knockout of CDYL in SY5Y cells led to a significant decrease of H3K27me3 and a moderate increase of H3K27ac at SCN8A intron region. Meanwhile, overexpression of CDYL in transgenic mice caused increase of H3K27me3 but the level of H3K27ac did not obviously change. We consider these results are mainly due to two reasons: first, from the technical

aspect, complete ablation of CDYL via CRISPR-Cas9 in a cell line could cause more dramatic effect on regional histone modification compared to mild overexpression of CDYL in experimental animals because of the dose-dependent effect; second, these experiments likely reflect that change of H3K27me3 at regional chromatin is the major underlying epigenetic mechanism for CDYL to repress SCN8A expression. To solidify this, we performed additional experiments demonstrating knockdown of EZH2 by specific siRNAs in SY5Y cells caused increased expression of SCN8A (Fig. 5I), and knockdown of EZH2 in the hippocampus of transgenic mice over-expressing CDYL by injection of lentivirus containing EZH2-shRNA alleviated CDYL-mediated changes of intrinsic excitability (Fig. 6B).

7. The age of transgenic mice and shRNA rats tested for seizure susceptibility is unclear. Authors should also clarify how long after the shRNA injections the rats were tested.

Authors: 6-8 weeks transgenic mice and their wild-type littermates were tested for seizure excitability. Since CDYL-shRNA-GFP was strongly expressed from 7 days post-infusion (DPI) and lasted at least 3 months (Fig. 2A-B), the tested mice were used at 14 days DPI. We have clarified the information in “Sterotaxically Guided CDYL-shRNA injection and EEG Recordings” section of Supplementary experimental procedures.

8. Sample sizes were not provided for some experiments.

Authors: We have clarified the sample sizes in each experiment in the revised manuscript.

9. Were different clonal populations of the SH-SY5Y CDYL-KO cells tested in order to confirm that the phenotype was not due to nonspecific CRISPR background editing?

Authors: We have tested different clonal populations of the SY5Y CDYL-KO cells and we observed similar effect of CDYL on SCN8A expression (Fig. 5E). The information regarding identification of different clonal populations of SY5Y CDYL-KO cells has been provided (Fig. 5E).

Minor concerns:

1. Source of the CamKII-Cre mice does not seem to be provided.

Authors: The CamKII-Cre mice were obtained from Jackson lab (B6.Cg-Tg(Camk2a-cre)T29-1Stl/J, JAX-005359). We have added this information in the method section.

2. The dose of kainic acid administered was not provided.

Authors: The dose of kainite acid used was 10mg/kg for rats (Shah et al. 2004) and 20mg/kg for mice (He et al. 2004, Huang et al. 2009). We have clarified this in the revised manuscript (line 15 to 16 page 29).

He XP, Kotloski R, Nef S, Luikart BW, Parada LF, McNamara JO. Conditional deletion of TrkB but not BDNF prevents epileptogenesis in the kindling model. *Neuron* 2004; 43:31–42.

Shah MM1, Anderson AE, Leung V, Lin X, Johnston D. Seizure-induced plasticity of h channels in entorhinal cortical layer III pyramidal neurons. *Neuron* 2004; 44(3):495-508.

Huang Z, Walker MC, Shah MM. Loss of dendritic HCN1 subunits enhances cortical excitability and epileptogenesis. *J Neurosci.* 2009; 29(35):10979-88.

3. Page 29, the reference given (Dudek et al. 2002) for the knockout of CDYL in SH-SY5Y cells seems incorrect.

Authors: Sorry for the mistake. We have replaced with the corrected reference.

Genome engineering using the CRISPR-Cas9 system. Ran FA, Hsu PD, Wright J, Agarwala V, Scott DA, Zhang F. *Nat Protoc.* 2013 Nov;8(11):2281-308

Genome-scale CRISPR-Cas9 knockout screening in human cells. Shalem O, Sanjana NE, Hartenian E, Shi X, Scott DA, Mikkelsen TS, Heckl D, Ebert BL, Root DE, Doench JG, Zhang F. *Science.* 2014 Jan 3;343(6166):84-7

4. Page 40, Figure 3 legend, ‘mice’s tails’ is incorrect grammar.

Authors: Sorry for the mistake, we have made the correction to “mice tails”. The relevant figure has been moved to supp. figure 4A.

5. Page 40, Figure 1 legend, ‘hyperpolarizing’ is spelt incorrectly.

Authors: We have corrected the spelling (Line 8 in Page 41).

6. Page 42, Figure 5 legend, does Part (A) and (B) correctly refer to mouse and rat data respectively?

Authors: Yes, Fig.5A and 5B refer to mouse and rat data respectively. We have clarified the issue in the relevant figure legend.

7. Figure 4A: The authors should state their rationale for searching +3 kb of the promoter. It is unclear how the authors defined “downstream” of the gene (polyA site, last coding exon, etc.).

Authors: +3 kb of the transcription start site was defined as the promoter according

to common criteria to analyze ChIP-seq data (Si et al. 2015). Searching +4 kb of the transcription start site as the promoter region yielded similar results as using +3 kb. “Downstream” is defined as distance from the transcription stop site of the gene. We have provided the information in the relevant part (Line 21 to 23 in Page 31).

Si W et al, Dysfunction of the reciprocal feedback loop between GATA3- and ZEB2-nucleated repression programs contributes to breast cancer metastasis. *Cancer Cell* 2015; 27(6):822-36.

8. *Figure 4C: The authors should provide access to the other genes identified by GO analysis and ChIP-Seq against CDYL.*

Authors: We have provided spreadsheet of the genes identified by GO analysis in Supplementary dataset (Supplementary Table 3). We have uploaded our CDYL ChIP-seq data to GEO, and the accession number is GSE96721.

9. *Figure 5C: TG4 does not appear to have knockdown of Nav1.6.*

Authors: TG4 actually exhibited knockdown of Nav1.6. We have replaced with a lighter exposure to show the effect more clearly in the revised manuscript (Fig. 5C).

Reviewer #3 (Remarks to the Author):

Chromodomain on Y-like (CDYL) is a REST corepressor that physically bridges REST and the histone methylase G9a to repress transcription. Using a combination of molecular, electrophysiology and ChIP-seq, Liu et al show that whereas neuronal activity suppresses CDYL, CDYL suppresses neuronal excitability. The authors further identify the mechanism by which this occurs. Whereas shRNA to CDYL increases firing frequency and decreases the threshold for cell firing, overexpression of CDYL in transgenic mice reduces firing frequency and increases the threshold for cell firing. The authors further show that pharmacological inhibition of axonal Nav1.6 in WT DGCs mimics the effects of CDYL overexpression in WT cells (increases the threshold for cell firing, with little or no effect on cells overexpressing CDYL. The authors propose a model whereby CDYL puts a brake on neuronal excitability by tempering the Nav1.6-mediated sodium current. In support of the model, shRNA to CDYL elicits a marked increase in the Nav1.6-mediated sodium current and 4,9-TTX, an inhibitor of Nav1.6, rescues the elevated sodium current observed in cells expressing CDYL shRNA. Using a kainic acid model of status epilepticus, the authors show that whereas shRNA to CDYL markedly increases seizure progression and severity, overexpression of CDYL slows them. Finally, the authors show that CDYL directly binds the SCN8A promoter. Moreover, in cells overexpressing CDYL, H3K27me3 (a mark of gene repression) is enriched, with little or no change in H3K27ac (a mark of

active gene transcription). In cells lacking CDYL, H3K27me3 is decreased, and H3K27ac is modestly increased. The authors conclude that CDYL-SCN8A axis affords a novel mechanism to regulate epileptogenesis and the neuronal hyperexcitability that induces seizures.

The present paper builds on a previous paper by several of the same authors that shows that CDYL is a negative regulator of dendritic arborization. The present paper is novel in that it is the first to address a role for the epigenetic factor CDYL in neuronal excitability and epileptogenesis. The experiments appear, for the most part, to be carefully executed. These positive aspects notwithstanding, there are several methodological and presentational aspects that should be addressed:

Major points:

1. Fig. 1: the quality of the representative Westerns shown in panels 1B, 5A and 5C are suboptimal and should be replaced.

Authors: We have replaced the Western blots in Fig. 1B, 5A, and 5C in the revision.

2. Fig. 1D-H: The authors show that three diverse stimuli known to increase network excitability, enriched environment, injection of NMDA or kainic acid, all decrease CDYL expression in the hippocampus. Many studies have shown that enriched environment reduces spontaneous seizures and neuronal death, and improve cognition in the face of epileptogenic seizures. Yet the authors report that overexpression of CDYL reduces seizure progression and severity. Given that enriched environment decreases CDYL expression, how do the authors interpret this apparent discrepancy?

Authors: We thank the reviewer for raising this point. The possible explanation is that in comparing with genetic manipulation of CDYL expression, environmental enrichment caused alteration in various signaling pathway and epigenetic proteins besides CDYL protein, such as ERK/MSK/ CREB pathway, ERK-MAPK signaling cascade and several histone acetyltransferases (HATs) (Sweatt JD, 2009). These epigenetic or genetic modifications may counterbalance with the influence of CDYL on epileptogenesis.

Sweatt JD. Experience-dependent epigenetic modifications in the central nervous system. *Biol Psychiatry*. 2009; 65(3):191-7.

3. Fig. 5: Authors show that SY5Y cells lacking CDYL exhibit a marked decrease in H3K27me3 and increase in H3K27ac enrichment at the SCN8A promoter. They conclude that CDYL inhibits SCN8A transcription by recruiting H3K27 methyltransferase and the epigenetic remodeling enzymes HDAC1 and 2 to an intronic

region of the gene and induce gene silencing. However, CDYL overexpression does not decrease H3K27ac enrichment. The authors should examine a potential causal relation between CDYL and histone modifications. Moreover, the authors cannot rule out the possibility that REST, a master repressor of neural genes that silences gene transcription via histone acetylation, or EZH2, a repressor that confers a trimethylation mark at H3K27, act as corepressors to silence SCN8A. In light of findings by other that CDYL is a REST corepressor that physically bridges REST and the histone methyltransferase G9a (Mulligan et al, 2008 Mol Cell), and that CDYL interacts with EZH2 directly to promote trimethylation of H3K27 at the promoters of target genes such as BDNF (Qi et al, J Neurosci, 2014), the author should perform additional experiments to examine a role for REST and EZH2 in repression of SCN8A.

Authors: We thank the reviewer's constructive comments. In Fig. 5G, knockout of CDYL in SY5Y cells led to a significant decrease of H3K27me3 and a moderate increase of H3K27ac at SCN8A intron region. Overexpression of CDYL in transgenic mice caused regional increase of H3K27me3 but the level of H3K27ac did not obviously change (Fig. 5D). We consider these results are mainly due to two reasons: first, from the technical aspect, complete ablation of CDYL via CRISPR-Cas9 in a cell line should cause more dramatic effect on regional histone modification compared to mild overexpression of CDYL in experimental animals because of the dose-dependent effect; second, these experiments indicate that change of H3K27me3 at regional chromatin is likely the major underlying epigenetic mechanism for CDYL to repress SCN8A expression. We have discussed this in the revision. Following the reviewer's suggestion, we performed additional experiments showing that knockdown of EZH2 in the hippocampus of transgenic mice over-expressing CDYL by injection of lentivirus containing EZH2-shRNA alleviated CDYL-mediated changes of intrinsic excitability (Fig. 6B). We also performed the experiments in SY5Y cells showing that knockdown of REST and EZH2 by specific siRNAs caused increased expression of SCN8A, an effect similar to CDYL depletion. The data are provided in Fig. 5I.

3. p22-23: Authors conclude that, "the CDYL-SCN8A axis is critical to regulate intrinsic plasticity of neuronal cells and targeting CDYL may provide a robust method of dampening hyperexcitability that leads to seizures." The paper would be substantially strengthened were the authors to examine a potential role for REST and/or EZH2 in these actions.

Authors: As stated above, we have provided experimental evidence showing that knockdown of EZH2 expression in the hippocampus of CDYL overexpressing mice by EZH2-shRNA lentiviral infection alleviated CDYL-mediated changes of intrinsic excitability (Fig. 6B). We have also reworded this sentence as suggested by reviewer #1.

4. The findings demonstrating the impact of shRNA-CDYL on Nav1.6 activity would

be further strengthened were the authors to determine whether overexpression of Nav1.6 could recapitulate the impact of CDYL knockdown on neuronal excitability.

Authors: We thank the reviewer's constructive comments. As the ORF of SCN8A is about 6kb, a length that is not suitable to generate high-titered lentivirus overexpressing SCN8A-GFP for animal injection, we performed two types of rescuing experiments to solidify the impact of shRNA-CDYL on Nav1.6 activity. First, we generated lentiviruses carrying shRNA-SCN8A. While injection of lentivirus carrying shRNA-CDYL in mice significantly lower neuronal AP threshold and facilitate the development of epilepsy (Fig. 2D and Fig. 8B-D), coinjection of shRNA-SCN8A and shRNA-CDYL to the mice cortical neurons almost completely abolished the AP threshold changes (Fig. 6D-E) and partially reversed the alteration in seizure susceptibility (Fig. 8H-I). In addition, we obtained SCN8A null heterozygote mice from our collaborator Prof. Yousheng Shu (Li et al., Plos Biology, 2014). These mice have reduced expression of SCN8A protein (Supplementary Fig. 5A) consistent with previously reports (Papale et al., Hum Mol Genet, 2009; Yin et al., Cerebral Cortex, 2015) and consequently exhibited more depolarized neuronal threshold (Supplementary Fig. 5B) and lower epilepsy susceptibility (Supplementary Fig. 8A-B) when compared with control mice. We demonstrated that SCN8A null heterozygotes injected with lentiviruses carrying CDYL-shRNA showed increased expression of SCN8A, recovered the AP threshold (Supplementary Fig. 5A-B), and partially reversed alteration in seizure susceptibility (Supplementary Fig. 8A-B). These data support SCN8A-encoded Nav1.6 sodium channels mainly underlie the CDYL-mediated functional changes on AP threshold and epilepsy susceptibility.

Li T et al, Action potential initiation in neocortical inhibitory interneurons. *PLoS Biol.* 2014; 9; 12(9): e1001944.

Papale et al, Heterozygous mutations of the voltage-gated sodium channel SCN8A are associated with spike-wave discharges and absence epilepsy in mice. *Hum Mol Genet.* 18(9): 1633-41, (2009).

Yin L et al, Selective Modulation of Axonal Sodium Channel Subtypes by 5-HT1A Receptor in Cortical Pyramidal Neuron. *Cereb Cortex.* 2015; pii: bhv245

REVIEWERS' COMMENTS:

Reviewer #1 (Remarks to the Author):

The authors have adequately responded to all of my comments.

Reviewer #2 (Remarks to the Author):

The authors have sufficiently addressed the concerns.

Reviewer #3 (Remarks to the Author):

The present paper is novel in that it is the first to address a role for the epigenetic factor CDYL in neuronal excitability and epileptogenesis. The authors have adequately addressed our concerns and the resulting manuscript is suitable for publication. These concerns include improving the quality of the westerns provided for Fig. 1 and Fig. 5. Also, they addressed the discrepancy between results from previous studies show that enriched environments can reduce spontaneous seizures and their findings that enriched environments decreased CDYL expression (leading to increased seizures. In addition, they performed additional experiments finding that they could alleviate CDYL-mediated changes in intrinsic excitability with knockdown of EZH2. Finally, they performed experiments to further support that Nav1.6 are responsible for CDYL-mediated functional changes on AP threshold and epilepsy susceptibility.